# Sharp Bounds for Generalized Causal Sensitivity Analysis

**Dennis Frauen, Valentyn Melnychuk & Stefan Feuerriegel**
LMU Munich
Munich Center for Machine Learning
{frauen,melnychuk,feuerriegel}@lmu.de

## Abstract

Causal inference from observational data is crucial for many disciplines such as medicine and economics. However, sharp bounds for causal effects under relaxations of the unconfoundedness assumption (causal sensitivity analysis) are subject to ongoing research. So far, works with sharp bounds are restricted to fairly simple settings (e.g., a single binary treatment). In this paper, we propose a unified framework for causal sensitivity analysis under unobserved confounding in various settings. For this, we propose a flexible generalization of the marginal sensitivity model (MSM) and then derive sharp bounds for a large class of causal effects. This includes (conditional) average treatment effects, effects for mediation analysis and path analysis, and distributional effects. Furthermore, our sensitivity model is applicable to discrete, continuous, and time-varying treatments. It allows us to interpret the partial identification problem under unobserved confounding as a distribution shift in the latent confounders while evaluating the causal effect of interest. In the special case of a single binary treatment, our bounds for (conditional) average treatment effects coincide with recent optimality results for causal sensitivity analysis. Finally, we propose a scalable algorithm to estimate our sharp bounds from observational data.

## 1 Introduction

Causal effects are crucial for decision-making in many disciplines, such as marketing [71], medicine [77], and economics [1]. For example, physicians need to know the treatment effects to personalize medical care, and governments are interested in the causal effects of policies on infection rates during a pandemic. In many such applications, randomized experiments are costly or infeasible, because of which causal effects must be estimated from observational data [59].

Estimating causal effects from observational data may lead to bias due to the existence of confounders, i.e., variables that affect both treatment and outcome [55]. A remedy is to observe all confounders and thus to assume unconfoundedness (e.g., as in [16, 63, 64]). However, in many practical applications, the assumption of unconfoundedness is violated. For example, electronic health records do not capture a patient's ethnic background, which is a known confounder in medicine [52]. In such cases, the causal effect is not identified from observational data, and unbiased estimation is thus impossible [27].

A popular way to perform causal inference in the presence of unobserved confounders is ***causal sensitivity analysis***. Causal sensitivity analysis aims to derive bounds on the causal effect of interest under relaxations of the unconfoundedness assumption. Here, the strength of unobserved confounding is typically controlled by some sensitivity parameter $\Gamma$, which also determines the tightness of the bounds. In practice, one chooses $\Gamma$ through domain knowledge [14, 33, 82] or data-driven heuristics [29]. Then, one derives that the causal effect of interest lies in some informative region. For example,

37th Conference on Neural Information Processing Systems (NeurIPS 2023).

a suitable $\Gamma$ may enable – despite unobserved confounding – to infer the sign of the treatment effect, which is often sufficient for consequential decision-making [35].

A common model for causal sensitivity analysis is the marginal sensitivity model (MSM) [31, 35, 36, 67]. A benefit of the MSM is that it does not impose any kind of parametric assumptions on the data-generating process. However, the standard MSM is only applicable in settings where the treatment is a single binary variable. Different extensions have been proposed for continuous treatments [10, 32, 45] and for time-varying treatments and confounders [10]. However, these works are restricted to specific settings, while a unified framework for causal sensitivity analysis is still missing.

**Contributions:**[1] In this paper, we propose a *generalized marginal sensitivity model (GMSM)*. Our GMSM provides a unified framework for causal sensitivity analysis under unobserved confounding in various settings with multiple discrete, continuous, and time-varying treatments. Crucially, our GMSM includes existing models, such as those in [67] and [32], as special cases. As a result, our GMSM enables a unified approach to deriving sharp bounds for a large class of

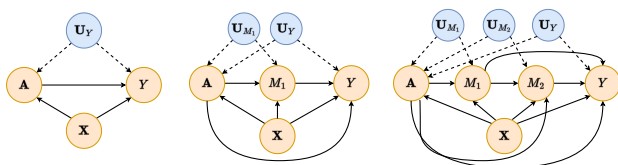

Figure 1: Three examples of causal inference settings where our GMSM and bounds are applicable. $M_1$, $M_2$ are mediators, $Y$ is the outcome, and $\mathbf{X}$ and $\mathbf{A}$ are (multiple) discrete, continuous, or time-varying covariates and treatments. Variables $\mathbf{U}_W$ are unobserved confounders between $\mathbf{A}$ and $W \in \{M_1, M_2, Y\}$.

causal effects. To do so, we bound a distribution shift in the unobserved confounders while performing an intervention on the treatments. As a result, we obtain sharp bounds for various causal effects, including (conditional) average treatment effects but also effects for mediation analysis and path analysis (see Fig. 1) and for distributional effects, which have not been studied under an MSM-type sensitivity analysis previously. We also show that, for binary treatments, our bounds coincide with recent optimality results for (conditional) average treatment effects under the MSM. Finally, we propose a scalable algorithm to estimate our sharp bounds from observational data and perform extensive computational experiments to show the validity of our bounds empirically.

## 2 Related work

In the following, we review related work on causal sensitivity analysis. For a more general review of partial identification and treatment effect estimation under unconfoundedness, we refer to Appendix A.

**Causal sensitivity analysis:** Causal sensitivity analysis dates back to a study from 1959 showing that unobserved confounders cannot explain away the causal effect of smoking on cancer [14]. This was formalized by introducing *sensitivity models* that yield bounds on the causal effect of interest under some restriction on the amount of confounding. Previous works introduced a variety of sensitivity models that make different assumptions about the data-generating process and the confounding mechanism. Examples include sensitivity models based on parametric assumptions [30, 62], difference between potential outcome distributions [60, 70], and Rosenbaum's sensitivity model that uses randomization tests [61].

**Marginal sensitivity model (MSM):** The *marginal sensitivity model (MSM)* [67] is a common model for sensitivity analysis aimed at settings with binary treatments. Many methods were proposed to estimate bounds from observational data under the MSM. Examples of such methods include linear fractional programs [82] and machine learning such as including kernel-based methods [35] and deep neural networks [31]. Recently, Dorn et al. [18] and Jin et al. [33] showed that estimates from the previous methods are too conservative bounds and, as a remedy, derived closed-form solutions for sharp bounds under the MSM. This was also extended to semiparametric inference [19, 53]. However, all previous methods are limited to binary treatments and (conditional) average treatment effects.

Different extensions for the MSM have been proposed. Jesson et al. [32], Bonvini et al. [10], and Marmarelis et al. [45] developed extensions for continuous treatments. Furthermore, Bonvini et al. [10] also extended the MSM to time-varying treatments and confounders. However, both works are *not* applicable to settings beyond the estimation of (conditional) average treatment effects, such as

---

[1]Code is available at https://github.com/DennisFrauen/SharpCausalSensitivity.

mediation analysis and path analysis or distributional effects. In contrast, we propose a generalized marginal sensitivity model that is compatible with binary, continuous, and time-varying treatments for a variety of causal effects.

**Research gap:** To the best of our knowledge, no existing causal sensitivity analysis based on the MSM provides a unified framework for deriving bounds for binary, continuous, and time-varying treatments. Furthermore, MSM-based causal sensitivity analysis is restricted to (conditional) average treatment effects and is thus *not* applicable for complex settings such as effects for mediation analysis and path analysis or distributional effects.

## 3 Generalized marginal sensitivity model (GMSM)

We first formally define a general setting for causal sensitivity analysis that includes mediation and path analysis (Sec. 3.1 and Sec. 3.2). We then propose our generalized marginal sensitivity model (GMSM) in Sec. 3.3 and compare it with existing sensitivity models from the literature.

**Notation:** We write random variables as capital letters $X$ and their realizations in lowercase $x$. Bold letters $\mathbf{X}$ or $\mathbf{x}$ represent (random) vectors. If $\mathbf{X} = (X_1, \ldots, X_\ell)$ is a sequence of random variables of length $\ell$, we denote $\bar{\mathbf{X}}_k = (X_1, \ldots, X_k)$ for $1 \leq k \leq \ell$. We denote probability distributions over $X$ as $\mathbb{P}^X$ where required. The probability mass function for a discrete $X$ is denoted as $\mathbb{P}(x) = \mathbb{P}(X = x)$. If $X$ is continuous, $\mathbb{P}(x)$ is the probability density function. We denote $\mathbb{P}(\cdot)$ as the corresponding probability mass/density function not evaluated at a specific $x$. Similarly, we write conditional probability mass functions/density functions as $\mathbb{P}(y \mid x) = \mathbb{P}(Y = y \mid X = x)$ and conditional expectations as $\mathbb{E}[Y \mid x] = \mathbb{E}[Y \mid X = x] = \int y \, \mathbb{P}(y \mid x) \, \mathrm{d}y$. We denote $\mathbb{P}(y \mid do(X = x))$ as the probability mass function/density function of $Y$ after performing the do-intervention $do(X = x)$ [55]. Finally, we define $\mathbf{X}_Y = pa(Y) \cap \mathbf{X}$, where $pa(Y)$ denotes the parents of $Y$ in a given causal graph.

### 3.1 Problem setup for generalized causal sensitivity analysis

We formalize causal sensitivity analysis based on Pearl's structural causal model framework [55].

**Definition 1.** A *structural causal model (SCM)* $\mathcal{M}$ is a tuple $(\mathbf{V}, \mathbf{U}, \mathcal{F}, \mathbb{P}^{\mathbf{U}})$, where $\mathbf{V} = (V_1, \ldots, V_k)$ are observed endogenous variables, $\mathbf{U}$ are unobserved exogenous variables determined outside of the model, $\mathcal{F} = \{f_{V_1}, \ldots, f_{V_k}\}$ is a set of functions so that each $f_{V_i}$ maps a set of parents $pa(V_i) \subseteq \mathbf{V} \cup \mathbf{U}$ to $V_i$, and $\mathbb{P}^{\mathbf{U}}$ is a probability distribution on $\mathbf{U}$.

Every SCM $\mathcal{M}$ induces unique directed graph[2] $\mathcal{G}_\mathcal{M}$ on $\mathbf{V} \cup \mathbf{U}$ by drawing a directed edge from $V_1$ to $V_2$ if $V_1 \in pa(V_2)$. In this paper, we assume that $\mathcal{G}_\mathcal{M}$ is acyclic, i.e., does not contain any directed cycle. Then, $\mathcal{M}$ induces a unique joint probability distribution $\mathbb{P}^{\mathbf{V} \cup \mathbf{U}}$ on $\mathbf{V} \cup \mathbf{U}$. Furthermore, $\mathcal{M}$ induces unique interventional distributions $\mathbb{P}^{\mathbf{V} \cup \mathbf{U}}_{do(\mathbf{A}=\mathbf{a})}$ when performing the do-intervention $do(\mathbf{A} = \mathbf{a})$ for some observed treatments $\mathbf{A} \subseteq \mathbf{V}$ [4, 55].

We consider settings where we observe four distinct types of endogenous variables $\mathbf{V} = \{\mathbf{X}, \mathbf{A}, \bar{\mathbf{M}}_\ell, Y\}$: observed confounders $\mathbf{X} \in \mathcal{X} \subseteq \mathbb{R}^{d_x}$, treatments $\mathbf{A} \in \mathcal{A} \subseteq \mathbb{R}^{d_a}$, discrete mediators $\bar{\mathbf{M}}_\ell = (M_1, \ldots, M_\ell)$ with $M_i \in \mathbb{N}$ for $1 \leq i \leq \ell$, and an outcome $Y \in \mathbb{R}$. In a medical setting, $\mathbf{X}$ might be patient characteristics (gender, age, medical history, etc.), $\mathbf{A}$ a medical treatment, $\bar{\mathbf{M}}_\ell$ a change in diet, and $Y$ a variable indicating a health outcome. Possible causal graphs are shown in Fig. 1. We assume w.l.o.g. that $(M_1, \ldots, M_\ell)$ are ordered causally, i.e., $\bar{\mathbf{M}}_{i-1}$ are parents of $M_i$ for each $i \in \{2, \ldots, \ell\}$. Given treatment interventions $\bar{\mathbf{a}}_{\ell+1} = (\mathbf{a}_1, \ldots, \mathbf{a}_{\ell+1})$, we are interested in causal effects of the form

$$Q(\mathbf{x}, \bar{\mathbf{a}}_{\ell+1}, \mathcal{M}) = \sum_{\bar{\mathbf{m}}_\ell \in \mathbb{N}^\ell} \mathcal{D}\left(\mathbb{P}^Y(\cdot \mid \mathbf{x}, \bar{\mathbf{m}}_\ell, do(\mathbf{A} = \mathbf{a}_{\ell+1}))\right) \prod_{i=1}^\ell \mathbb{P}(m_i \mid \mathbf{x}, \bar{\mathbf{m}}_{i-1}, do(\mathbf{A} = \mathbf{a}_i)),$$
(1)

where $\mathcal{D}$ is some functional that maps the density $\mathbb{P}^Y(\cdot \mid \mathbf{x}, \mathbf{m}, do(\mathbf{A} = \mathbf{a}_{\ell+1}))$ to a scalar value and we sum over all possible realizations $\bar{\mathbf{m}}_\ell$ of $\bar{\mathbf{M}}_\ell$. We are also interested in averaged causal effects

---

[2]Note that we consider graphs over both observed and unobserved variables. In the causal inference literature, these are also called augmented causal graphs[9].

$\int_{\mathcal{X}} Q(\mathbf{x}, \bar{\mathbf{a}}_{\ell+1}, \mathcal{M}) \, \mathbb{P}(\mathbf{x}) \, \mathrm{d}\mathbf{x}$ or differences (Appendix D). The effect $Q(\mathbf{x}, \bar{\mathbf{a}}_{\ell+1}, \mathcal{M})$ generalizes many common effects across different causal inference settings.

**Example 1** (CATE, $\ell = 0$). If $\mathcal{D} = \mathbb{E}[\cdot]$ is the expectation functional, Eq. (1) reduces to $Q(\mathbf{x}, \mathbf{a}, \mathcal{M}) = \mathbb{E}[Y \mid \mathbf{x}, do(\mathbf{A} = \mathbf{a})]$. When $\mathbf{A}$ is continuous, this is known as the conditional dose-response function [7, 32]. For binary treatments $\mathbf{A} \in \{0, 1\}$, the query $Q(\mathbf{x}, 1, \mathcal{M}) - Q(\mathbf{x}, 0, \mathcal{M})$ is known as the conditional average treatment effect (CATE) [16, 78], and its averaged version as the average treatment effect (ATE) [64, 69].

**Example 2** (Mediation analysis, $\ell = 1$). If $\mathcal{D} = \mathbb{E}[\cdot]$ and $\bar{\mathbf{M}}_1 = M$ is a single mediator, Eq. (1) reduces to $Q(\mathbf{x}, \bar{\mathbf{a}}_1, \mathcal{M}) = \sum_m \mathbb{E}[Y \mid \mathbf{x}, m, do(\mathbf{A} = \mathbf{a}_2)] \, \mathbb{P}(m \mid \mathbf{x}, do(\mathbf{A} = \mathbf{a}_1))$. If $A \in \{0, 1\}$ is binary and the $M$–$Y$ relationship is unconfounded, we obtain the following causal effects studied in mediation analysis [56]: $Q(\mathbf{x}, (\mathbf{a}_1 = 0, \mathbf{a}_2 = 1), \mathcal{M}) - Q(\mathbf{x}, (\mathbf{a}_1 = 0, \mathbf{a}_2 = 0), \mathcal{M})$ is the (conditional) natural direct effect (NDE), and $Q(\mathbf{x}, (\mathbf{a}_1 = 1, \mathbf{a}_2 = 0), \mathcal{M}) - Q(\mathbf{x}, (\mathbf{a}_1 = 0, \mathbf{a}_2 = 0), \mathcal{M})$ is the (conditional) natural indirect effect (NIE).

In general, Eq. (1) includes so-called *path-specific effects* if the relationship between mediators and outcome is unconfounded [2, 15]. For example, by setting $\mathbf{a}_1 = 1$ and $\mathbf{a}_k = 0$ for all $2 \leq k \leq \ell + 1$, we obtain the indirect effect that is passed through the mediator sequence $(M_1, \ldots, M_\ell)$. Path-specific effects are important in various applications, including algorithmic fairness, where the aim is to mitigate effects through paths that are considered unfair [50, 51]. Furthermore, we can set $\mathcal{D}$ to a quantile instead of using the mean in all the examples above. This results in *distributional* versions of CATE, NDE, NIE, and path-specific effects. For example, if the outcome distribution is skewed or contains outliers, practitioners might prefer the median or other quantiles over the mean due to their robustness properties [17].

If all confounders between treatment and mediators and outcome were observed, we could (under additional assumptions) identify the causal effect $Q(\mathbf{x}, \bar{\mathbf{a}}_{\ell+1}, \mathcal{M})$ from the observational distribution $\mathbb{P}^{\mathbf{V}}$ on $\mathbf{V}$ by replacing all *do*-operations with conditional probabilities according to the backdoor criterion [55]. However, under unobserved confounding, identifiability is not possible because SCMs $\mathcal{M}$ and $\mathcal{M}'$ exist, which induce the same observational distribution $\mathbb{P}^{\mathbf{V}}$, but which result in different causal effects $Q(\mathbf{x}, \bar{\mathbf{a}}_{\ell+1}, \mathcal{M}) \neq Q(\mathbf{x}, \bar{\mathbf{a}}_{\ell+1}, \mathcal{M}')$ [54, 74].

### 3.2 Causal sensitivity analysis

In the following, we formalize causal sensitivity analysis as maximizing/minimizing the causal effect $Q(\mathbf{x}, \bar{\mathbf{a}}_{\ell+1}, \mathcal{M})$ over all SCMs $\mathcal{M}$ that are compatible with a predefined *sensitivity model*. Similar approaches have been used for partial identification with instrumental variables [40, 54] and testing identifiability of counterfactuals [74, 75], where all SCMs are considered that are compatible with the observational data. However, a sensitivity model must additionally restrict the joint distribution of both observed and unobserved variables in order to allow for informative bounds on the causal effect.

**Definition 2** (Sensitivity model). Let $\mathbb{P}^{\mathbf{V}}$ denote the distribution of the observed variables $\mathbf{V} = (\mathbf{X}, \mathbf{A}, \bar{\mathbf{M}}_\ell, Y)$. A *sensitivity model* $\mathcal{S}$ is a tuple $(\mathbf{U}, \mathcal{P})$, where $\mathbf{U} = (\mathbf{U}_W)_W$ are unobserved confounders between $\mathbf{A}$ and $W \in \{M_1, \ldots, M_\ell, Y\}$, respectively (see Fig. 1) and a family $\mathcal{P}$ of joint probability distributions on $\mathbf{V} \cup \mathbf{U}$, such that, for all $\mathbb{P} \in \mathcal{P}$, it holds that $\int \mathbb{P}(\mathbf{v}, \mathbf{u}) \, \mathrm{d}\mathbf{u} = \mathbb{P}^{\mathbf{V}}(\mathbf{v})$. We denote the set of all SCMs $\mathcal{M}$ *compatible* with $\mathcal{S}$ (i.e., that respect the causal graph, induce a distribution $\mathbb{P} \in \mathcal{P}$, and do not contain additional confounders) as $\mathcal{C}(\mathcal{S})$ (see Appendix E).

Our definition of sensitivity models excludes unobserved confounding between mediators and the outcome. This ensures that the causal effect from Eq. (1) can be interpreted as a path-specific effect [2, 56]. We refer to Sec. 6 for a detailed discussion.

Using the definitions above, we can define *causal sensitivity analysis* as the following partial identification problem: we aim to obtain bounds $Q^-(\mathbf{x}, \bar{\mathbf{a}}_{\ell+1}, \mathcal{S}) \leq Q^+(\mathbf{x}, \bar{\mathbf{a}}_{\ell+1}, \mathcal{S})$ so that

$$Q^+(\mathbf{x}, \bar{\mathbf{a}}_{\ell+1}, \mathcal{S}) = \sup_{\mathcal{M} \in \mathcal{C}(\mathcal{S})} Q(\mathbf{x}, \bar{\mathbf{a}}_{\ell+1}, \mathcal{M}) \text{ and } Q^-(\mathbf{x}, \bar{\mathbf{a}}_{\ell+1}, \mathcal{S}) = \inf_{\mathcal{M} \in \mathcal{C}(\mathcal{S})} Q(\mathbf{x}, \bar{\mathbf{a}}_{\ell+1}, \mathcal{M}). \quad (2)$$

$Q^+(\mathbf{x}, \bar{\mathbf{a}}_{\ell+1}, \mathcal{S})$ is the maximal causal effect that can be achieved by any SCM that is compatible with the sensitivity mode $\mathcal{S}$ (and vice versa for $Q^-(\mathbf{x}, \bar{\mathbf{a}}_{\ell+1}, \mathcal{S})$). Hence, if the sensitivity model is valid, i.e., contains the ground-truth distribution over observed variables and unobserved confounders, we know that the ground-truth causal effect must be contained in the interval $[Q^-(\mathbf{x}, \bar{\mathbf{a}}_{\ell+1}, \mathcal{S}), Q^+(\mathbf{x}, \bar{\mathbf{a}}_{\ell+1}, \mathcal{S})]$. Bounds for average causal effects and effect differences follow immediately (see Appendix D).

## 3.3 Generalized marginal sensitivity model (GMSM)

We introduce now the GMSM. We begin by providing the general definition and then show that this extends various marginal sensitivity models from existing literature.

**Definition 3** (GMSM). The *generalized marginal sensitivity model (GMSM)* is a sensitivity model $(\mathbf{U}, \mathcal{P})$, where $\mathcal{P}$ contains all $\mathbb{P}$ that satisfy the following sensitivity constraint: For each $W \in \{M_1, \ldots, M_\ell, Y\}$, there exist bounds $s_W^-(\mathbf{a}, \mathbf{x}) \leq 1 \leq s_W^+(\mathbf{a}, \mathbf{x})$, so that, for all $\mathbf{u}_W$, $\mathbf{x}$, and $\mathbf{a}$

$$s_W^-(\mathbf{a}, \mathbf{x}) \leq \frac{\mathbb{P}(\mathbf{U}_W = \mathbf{u}_W \mid \mathbf{x}, \mathbf{a})}{\mathbb{P}(\mathbf{U}_W = \mathbf{u}_W \mid \mathbf{x}, do(\mathbf{A} = \mathbf{a}))} \leq s_W^+(\mathbf{a}, \mathbf{x}). \tag{3}$$

The GMSM bounds the distribution shift in the unobserved confounders $\mathbf{U}_W$ when performing the intervention $do(\mathbf{A} = \mathbf{a})$ instead of conditioning on $\mathbf{A} = \mathbf{a}$. This is a restriction on the strength of the effect the unobserved confounders $\mathbf{U}_W$ can have on the treatment $\mathbf{A}$. If $\mathbf{U}_W$ has no effect on $\mathbf{A}$, Eq. (3) holds with $s_W^-(\mathbf{a}, \mathbf{x}) = s_W^+(\mathbf{a}, \mathbf{x}) = 1$. Hence, the further $s_W^-(\mathbf{a}, \mathbf{x})$ and $s_W^+(\mathbf{a}, \mathbf{x})$ deviate from 1, the larger is the effect from $\mathbf{U}_W$ on $\mathbf{A}$ that the GMSM allows for. We will often use a *weighted* GMSM, which expresses the bounds in terms of a sensitivity parameter.

**Definition 4** (Weighted GMSM). A *weighted GMSM* is a GMSM where $s_W^-(\mathbf{a}, \mathbf{x})$ and $s_W^+(\mathbf{a}, \mathbf{x})$ can be written as $s_W^-(\mathbf{a}, \mathbf{x}) = \frac{1}{(1-\Gamma_W)q_W(\mathbf{a}, \mathbf{x}) + \Gamma_W}$ and $s_W^+(\mathbf{a}, \mathbf{x}) = \frac{1}{(1-\Gamma_W^{-1})q_W(\mathbf{a}, \mathbf{x}) + \Gamma_W^{-1}}$ for a sensitivity parameter $\Gamma_W \geq 1$ and a weight function $q_W(\mathbf{a}, \mathbf{x}) \in [0, 1]$ for all $\mathbf{x}$ and $\mathbf{a}$.

In a weighted GMSM, the sensitivity parameter $\Gamma_W$ captures the overall restriction on the unobserved confounding strength across individuals. If $\Gamma_W = 1$, no unobserved confounding is allowed, and unconfoundedness holds. For $\Gamma_W \to \infty$, the restriction is relaxed completely, and arbitrary confounding strength is allowed. The weight function $q_W(\mathbf{a}, \mathbf{x}) \in [0, 1]$ offers several advantages. It allows to further restrict confounding for individuals with treatments $\mathbf{a}$ and covariates $\mathbf{x}$. As $q_W(\mathbf{a}, \mathbf{x}) \to 1$, confounding is more strongly restricted until unconfoundedness is reached. This is helpful in applications where prior knowledge about the confounding structure is available. As an example, consider a medical setting where we want to estimate the effect of new drug treatments on the risk of developing a certain disease, but we suspect that the data is confounded by the individual's genetic risk for the disease, which is not measured in the observational data. However, we also might know that genetic risk for the disease is not relevant for a specific combination of age and gender.

**Comparison with the MSM and its extensions:** In the following, we show that the weighted GMSM extends popular sensitivity models from the literature. Proofs are in Appendix C. Note that existing sensitivity models do not consider settings with mediators (i.e., $\ell = 0$), we thus can write $\Gamma = \Gamma_Y$ for the sensitivity parameter, $q(\mathbf{a}, \mathbf{x}) = q_Y(\mathbf{a}, \mathbf{x})$ for the weight function, and $\mathbf{U} = \mathbf{U}_Y$ for the unobserved confounders. For binary treatments $\mathbf{A} = A \in \{0, 1\}$, the marginal sensitivity model (MSM) [67] is defined via $\frac{1}{\Gamma} \leq \frac{\pi(\mathbf{x})}{1-\pi(\mathbf{x})} \frac{1-\pi(\mathbf{x}, \mathbf{u})}{\pi(\mathbf{x}, \mathbf{u})} \leq \Gamma$, where $\pi(\mathbf{x}) = \mathbb{P}(A = 1 \mid \mathbf{x})$ denotes the observed propensity score and $\pi(\mathbf{x}, \mathbf{u}) = \mathbb{P}(A = 1 \mid \mathbf{x}, \mathbf{u})$ denotes the full propensity score. For continuous treatments $\mathbf{A}$, the continuous marginal sensitivity model (CMSM) [32] is defined via $\frac{1}{\Gamma} \leq \frac{\mathbb{P}(\mathbf{a} \mid \mathbf{x}, \mathbf{u})}{\mathbb{P}(\mathbf{a} \mid \mathbf{x})} \leq \Gamma$. For longitudinal settings with time-varying observed confounders $\mathbf{X} = (\mathbf{X}_1, \ldots, \mathbf{X}_T)$, unobserved confounders $\mathbf{U} = (\mathbf{U}_1, \ldots, \mathbf{U}_T)$, treatments $\mathbf{A} = (\mathbf{A}_1, \ldots, \mathbf{A}_T)$, we define the longitudinal marginal sensitivity model (LMSM) via $\frac{1}{\Gamma} \leq \prod_{t=1}^{T} \frac{\mathbb{P}(\mathbf{a}_t \mid \bar{\mathbf{x}}_T, \bar{\mathbf{u}}_t, \bar{\mathbf{a}}_{t-1})}{\mathbb{P}(\mathbf{a}_t \mid \bar{\mathbf{x}}_T, \bar{\mathbf{a}}_{t-1})} \leq \Gamma$.

**Lemma 1.** *The MSM, CMSM and LMSM are special cases of the weighted GMSM by choosing the weight functions $q(\mathbf{a}, \mathbf{x}) = \mathbb{P}(\mathbf{a} \mid \mathbf{x})$ (MSM) and $q(\mathbf{a}, \mathbf{x}) = 0$ (CMSM, LMSM).*

Lemma 1 provides a *new interpretation of the MSM*: If the probability $\mathbb{P}(\mathbf{a} \mid \mathbf{x})$ is large, the MSM restricts the confounding strength because most of the randomness in the treatment is already explained by the observed confounders $\mathbf{x}$. We can extend this approach to arbitrary discrete treatments within the weighted GMSM framework. The CMSM and LMSM apply the same confounding restriction for all $\mathbf{a}$ and $\mathbf{x}$.

## 4 Bounding causal effects under the GMSM

In this section, we derive sharp bounds under our GMSM. We first quantify the maximal shift in conditional distributions (Sec. 4.1). This allows us to derive an algorithm to compute explicit bounds (Sec. 4.2). Finally, we show how these bounds can be estimated from finite data (Sec. 4.3).

## 4.1 Shifting interventional distributions

In the following, we provide some intuition before stating the main result. Let us consider a simple setting with treatment $\mathbf{A}$, a single unobserved confounder $\mathbf{U}_Y = U \in \mathbb{R}$, outcome $Y$, and GMSM bounds $s_Y^-(\mathbf{a})$ and $s_Y^+(\mathbf{a})$ (see Fig. 2, left). Any SCM $\mathcal{M}$ that is compatible with $\mathcal{S}$ describes the relationship between $\mathbf{A}$, $U$, and $Y$ via a functional assignment $Y = f_Y(\mathbf{A}, U)$. For a fixed treatment $\mathbf{a}$, we denote $f_Y(\mathbf{a}, \cdot)$ as $f_\mathbf{a}$. We are interested in the *interventional density* $\mathbb{P}(y \mid do(\mathbf{A} = \mathbf{a})) = \int \mathbb{P}(y \mid \mathbf{a}, u)\, \mathbb{P}(u)\, \mathrm{d}u = f_{\mathbf{a}\#}\mathbb{P}^U(y)$, where $f_{\mathbf{a}\#}\mathbb{P}^U$ denotes the push-forward distribution induced by $f_\mathbf{a}$ on $\mathbb{P}^U$. However, we only have access to the *observational density* $\mathbb{P}(y \mid \mathbf{a}) = \int \mathbb{P}(y \mid \mathbf{a}, u)\, \mathbb{P}(u \mid \mathbf{a})\, \mathrm{d}u = f_{\mathbf{a}\#}\mathbb{P}^{U|\mathbf{a}}(y)$. Hence, for a fixed functional assignment $f_\mathbf{a}$, we can quantify the discrepancy between $\mathbb{P}(y \mid do(\mathbf{A} = \mathbf{a}))$ and $\mathbb{P}(y \mid \mathbf{a})$ via the distribution shift between $\mathbb{P}^U$ and $\mathbb{P}^{U|\mathbf{a}}$.

Fig. 2 shows a toy example where $\mathbb{P}^{U|\mathbf{a}}$ is the uniform distribution on $[0, 1]$, and the functional assignment $f_\mathbf{a}$ is the inverse standard normal CDF $\Phi^{-1}$, so that $\mathbb{P}(y \mid \mathbf{a})$ is the standard normal probability density. We now want to "right-shift" the interventional density $\mathbb{P}(y \mid do(\mathbf{A} = \mathbf{a}))$ as much as possible, so that $F(y \mid a) \gg F(y \mid do(\mathbf{A} = \mathbf{a}))$ for the CDFs. To achieve this, the distribution $\mathbb{P}^U$ must put more probability mass on the right-hand side of the unit interval $[0, 1]$ as compared to $\mathbb{P}^{U|\mathbf{a}}$ (see Fig. 2, left).

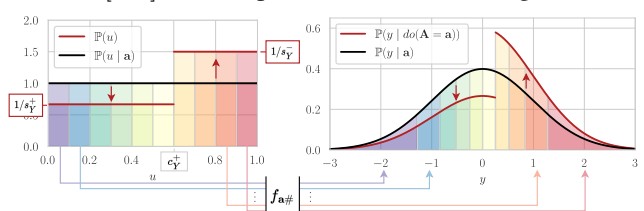

Then, the functional assignment will also push more probability mass to the right of $\mathbb{P}(y \mid do(\mathbf{A} = \mathbf{a}))$ as compared to $\mathbb{P}(y \mid \mathbf{a})$ (see Fig. 2, right). However, the GMSM bounds the distribution shift between $\mathbb{P}^U$ and $\mathbb{P}^{U|\mathbf{a}}$ via $1/s_Y^+(\mathbf{a}) \leq \mathbb{P}(u) \leq 1/s_Y^-(\mathbf{a})$ (Eq. (3) using that $\mathbb{P}(u) = \mathbb{P}(u \mid do(\mathbf{A} = \mathbf{a}))$). Intuitively, the maximal possible right shift under the

Figure 2: Intuition for bounding interventional distributions under our GMSM.

GMSM should occur by choosing $\mathbb{P}(u)$ so that the bounds are attained, i.e., $\mathbb{P}(u) = \mathbb{1}(u \leq c_Y^+)(1/s_Y^+) + \mathbb{1}(u > c_Y^+)(1/s_Y^-)$ for some $c_Y^+$ that can be obtained via the normalization constraint $\int \mathbb{P}(u)\, \mathrm{d}u = 1$. The corresponding "right-shifted" interventional density $\mathbb{P}_+(y \mid do(\mathbf{A} = \mathbf{a}))$ is defined via the push-forward (see Fig. 2).

It turns out that the density $\mathbb{P}_+(y \mid do(\mathbf{A} = \mathbf{a}))$ is the *maximally right-shifted* interventional distribution that can be obtained under any SCM that is compatible with the GMSM. Furthermore, the above arguments can be generalized beyond the toy example from Fig. 2.

**Theorem 1.** *Let $\mathcal{S}$ be a GMSM with bounds $s_W^- = s_W^-(\mathbf{a}, \mathbf{x})$ and $s_W^+ = s_W^+(\mathbf{a}, \mathbf{x})$ for $W \in \{M_1, \dots, M_\ell, Y\}$. We define $c_W^+ = \frac{(1 - s_W^-)s_W^+}{s_W^+ - s_W^-}$. If $W \in \mathbb{R}$ is continuous, we define the probability density function*

$$\mathbb{P}_+(w \mid \mathbf{x}, \mathbf{m}_W, \mathbf{a}) = \begin{cases} (1/s_W^+)\, \mathbb{P}(w \mid \mathbf{x}, \mathbf{m}_W, \mathbf{a}), & \text{if } F(w) \leq c_W^+, \\ (1/s_W^-)\, \mathbb{P}(w \mid \mathbf{x}, \mathbf{m}_W, \mathbf{a}), & \text{if } F(w) > c_W^+, \end{cases} \quad (4)$$

*where $F(\cdot)$ is the CDF corresponding to $\mathbb{P}(\cdot \mid \mathbf{x}, \mathbf{m}_W, \mathbf{a})$. If $W \in \mathbb{N}$ is discrete, we define the probability mass function*

$$\mathbb{P}_+(w \mid \mathbf{x}, \mathbf{m}_W, \mathbf{a}) = \begin{cases} (1/s_W^+)\, \mathbb{P}(w \mid \mathbf{x}, \mathbf{m}_W, \mathbf{a}), & \text{if } F(w) < c_W^+, \\ (1/s_W^-)\, \mathbb{P}(w \mid \mathbf{x}, \mathbf{m}_W, \mathbf{a}), & \text{if } F(w-1) > c_W^+, \\ (1/s_W^+)\left(c_W^+ - F(w-1)\right) + (1/s_W^-)\left(F(w) - c_W^+\right), & \text{else.} \end{cases} \quad (5)$$

*Let $F_+(\cdot)$ denote the conditional CDF corresponding to $\mathbb{P}_+(\cdot \mid \mathbf{x}, \mathbf{m}_W, \mathbf{a})$ and let $F_-(\cdot)$ be the conditional CDF of $\mathbb{P}_-(\cdot \mid \mathbf{x}, \mathbf{m}_W, \mathbf{a})$, which is defined by swapping signs (in $c_W$ and $s_W$). For any SCM $\mathcal{M}$, denote the CDF corresponding to $\mathbb{P}_\mathcal{M}(\cdot \mid \mathbf{x}, \mathbf{m}_W, do(\mathbf{A} = \mathbf{a}))$ as $F_\mathcal{M}(\cdot)$. Then, for all $w$,*

$$F_+(w) \leq \inf_{\mathcal{M} \in \mathcal{C}(\mathcal{S})} F_\mathcal{M}(w) \quad \text{and} \quad F_-(w) \geq \sup_{\mathcal{M} \in \mathcal{C}(\mathcal{S})} F_\mathcal{M}(w). \quad (6)$$

*Assume now that $\mathbb{P}(\mathbf{u}_W \mid \mathbf{x}, do(\mathbf{A} = \mathbf{a})) = \mathbb{P}(\mathbf{u}_W \mid \mathbf{x})$. Then, the bounds are sharp (i.e., equality holds in Eq. (6)) whenever $\mathbf{A}$ is discrete and it holds that $1/s_W^+ \geq \mathbb{P}(\mathbf{a} \mid \mathbf{x})$, or $\mathbf{A}$ is continuous.*

*Proof.* See Appendix B. ∎

The result in Theorem 1 does not depend on the distribution or dimensionality of unobserved confounders, and it does not depend on any specific SCM. As such, our sharp bounds are applicable to a wide class of causal effects, without restricting assumptions on the confounding structure beyond the sensitivity constraint of the GMSM.

In case $\mathcal{S}$ is a weighted GMSM with sensitivity parameters $\Gamma_W$ for all $W \in \{M_1, \ldots, M_\ell, Y\}$, the quantiles $c_W^+$ and $c_W^-$ are of the particular simple form $c_W^+ = \Gamma_W/(1 + \Gamma_W)$ and $c_W^- = 1/(1 + \Gamma_W)$. Furthermore, the discrete sharpness condition simplifies to $(1 - \Gamma_W^{-1})q_W(\mathbf{a}, \mathbf{x}) + \Gamma_W^{-1} \geq \mathbb{P}(\mathbf{a} \mid \mathbf{x})$. In particular, the bounds are sharp whenever we choose a weighting function that satisfies $q_W(\mathbf{a}, \mathbf{x}) \geq \mathbb{P}(\mathbf{a} \mid \mathbf{x})$. The assumption $\mathbb{P}(\mathbf{u}_W \mid \mathbf{x}, do(\mathbf{A} = \mathbf{a})) = \mathbb{P}(\mathbf{u}_W \mid \mathbf{x})$ excludes the time-varying case (LMSM). Deriving sharp bounds for the LMSM is an interesting direction for future research.

## 4.2 Bounding causal effects

We now leverage Theorem 1 to obtain explicit solutions for the partial identification problem from Eq. (2) with *monotone* $\mathcal{D}$ (see Appendix B). This includes expectation and distributional effects.

**Corollary 1** (Bounds without mediators). *If $\ell = 0$ and $\mathcal{D}$ is monotone, we obtain sharp bounds*

$$Q^+(\mathbf{x}, \mathbf{a}, \mathcal{S}) \leq \mathcal{D}\left(\mathbb{P}_+^Y(\cdot \mid \mathbf{x}, \mathbf{a})\right) \quad and \quad Q^-(\mathbf{x}, \mathbf{a}, \mathcal{S}) \geq \mathcal{D}\left(\mathbb{P}_-^Y(\cdot \mid \mathbf{x}, \mathbf{a})\right), \tag{7}$$

*and sharpness holds under the same conditions as in Theorem 1.*

*Proof.* See Appendix B. ∎

---

**Algorithm 1:** Causal sensitivity analysis with mediators

**Input** : Causal query $Q(\mathbf{x}, \bar{\mathbf{a}}_{\ell+1}, \mathcal{M})$, GMSM $\mathcal{S}$ with $s_W^+$ and $s_W^-$.
**Output:** Upper bound $Q^+(\mathbf{x}, \bar{\mathbf{a}}_{\ell+1}, \mathcal{S})$
`// Outcome bound`
$c_W^+ \leftarrow \frac{(1 - s_W^-)s_W^+}{s_W^+ - s_W^-}$ for $W \in \{M_1, \ldots, M_\ell, Y\}$
$Q_{\ell+1}^+(\bar{\mathbf{m}}_\ell) \leftarrow \mathcal{D}\left(\mathbb{P}_+^Y(\cdot \mid \mathbf{x}, \bar{\mathbf{m}}_\ell, \mathbf{a}_{\ell+1})\right)$ for $\bar{\mathbf{m}}_\ell \in supp(\bar{\mathbf{M}}_\ell)$
`// Adjusting for confounding in mediators`
**for** $i \in \{\ell, \ldots, 1\}$ **do**
  **for** $\bar{\mathbf{m}}_{i-1} \in supp(\bar{\mathbf{M}}_{i-1})$ **do**
    $\pi \leftarrow$ Permutation map in ascending order of
    $\left(Q_{i+1}^+(\bar{\mathbf{m}}_{i-1}, \pi(m_i))\right)_{m_i \in supp(M_i)}$
    $\widetilde{F}(m_i) \leftarrow \sum_{m:\pi(m) \leq m_i} \mathbb{P}(M_i = m \mid \mathbf{x}, \bar{\mathbf{m}}_{i-1}, \mathbf{a}_i)$

    $\mathbb{P}_+(m_i) \leftarrow \begin{cases} (1/s_{M_i}^+)\mathbb{P}(m_i \mid \mathbf{x}, \bar{\mathbf{m}}_{i-1}, \mathbf{a}_i), \\ \quad \text{if } \widetilde{F}(\pi(m_i)) < c_{M_i}^+, \\ (1/s_{M_i}^-)\mathbb{P}(m_i \mid \mathbf{x}, \bar{\mathbf{m}}_{i-1}, \mathbf{a}_i), \\ \quad \text{if } \widetilde{F}(\pi(m_i) - 1) > c_{M_i}^+, \\ (1/s_{M_i}^+)\left(c_{M_i}^+ - \widetilde{F}(\pi(m_i) - 1)\right) \\ + (1/s_{M_i}^-)\left(\widetilde{F}(\pi(m_i)) - c_{M_i}^+\right), \\ \quad \text{else.} \end{cases}$

    $Q_i^+(\bar{\mathbf{m}}_{i-1}) \leftarrow \sum_{m_i} Q_{i+1}^+(\bar{\mathbf{m}}_{i-1}, m_i) \mathbb{P}_+(m_i)$
  **end**
**end**
$Q^+(\mathbf{x}, \bar{\mathbf{a}}_{\ell+1}, \mathcal{S}) \leftarrow Q_1^+$

---

If $\mathcal{D}$ is the expectation functional, $\mathbf{A}$ is binary, and $Y$ is continuous, this coincides with the optimality result from Dorn and Guo [18]. Hence, Corollary 1 generalizes the result from Dorn and Guo [18] to distributional effects and arbitrary treatments (e.g., categorical, continuous, or time-varying). In their paper, Dorn and Guo proved the sharpness of the bounds by using the Neyman-Pearson Lemma. In contrast, we take the more principled approach outlined in Sec. 4.1, which is applicable to more general settings.

In the following, we consider settings with mediators, i.e., we aim to derive bounds for the causal effect in Eq. (1). The idea is to first obtain outcome bounds $\mathcal{D}\left(\mathbb{P}_+^Y(\cdot \mid \mathbf{x}, \bar{\mathbf{m}}_\ell, \mathbf{a}_{\ell+1})\right)$ conditioned on all possible values $\bar{\mathbf{m}}_\ell \in supp(\bar{\mathbf{M}}_\ell)$ in the support of the mediators $\bar{\mathbf{M}}_\ell$. Without unobserved confounding between treatments and mediators, we can obtain the upper bound $Q^+(\mathbf{x}, \bar{\mathbf{a}}_{\ell+1}, \mathcal{S})$ from Eq. (1) by replacing $\mathcal{D}\left(\mathbb{P}^Y(\cdot \mid \mathbf{x}, \bar{\mathbf{m}}_\ell, do(\mathbf{A} = \mathbf{a}_{\ell+1}))\right)$ with $\mathcal{D}\left(\mathbb{P}_+^Y(\cdot \mid \mathbf{x}, \bar{\mathbf{m}}_\ell, \mathbf{a}_{\ell+1})\right)$. With unobserved confounding, we need to additionally take the distribution shift in the mediators into account. To maximize the causal effect, the shifted mediator distribution should put more probability mass on values $\bar{\mathbf{m}}_\ell$ for which $\mathcal{D}\left(\mathbb{P}_+^Y(\cdot \mid \mathbf{x}, \bar{\mathbf{m}}_\ell, \mathbf{a}_{\ell+1})\right)$ is large. Hence, we can apply the maximal right-shift from Theorem 1 to the mediator distributions in Eq. (1), but where the values $\bar{\mathbf{m}}_\ell$ are permuted to order $\mathcal{D}\left(\mathbb{P}_+^Y(\cdot \mid \mathbf{x}, \bar{\mathbf{m}}_\ell, \mathbf{a}_{\ell+1})\right)$ in ascending order. We provide the details on our iterative procedure to compute the upper bound $Q^+(\mathbf{x}, \bar{\mathbf{a}}_{\ell+1}, \mathcal{S})$ in Algorithm 1. The lower bound $Q^-(\mathbf{x}, \bar{\mathbf{a}}_{\ell+1}, \mathcal{S})$ can be computed analogously by swapping signs in $c_W$ and $s_W$.

**Corollary 2.** *Under the Assumptions of Theorem 1, Algorithm 1 returns the sharp bounds from Eq. (2).*

*Proof.* See Appendix B. ∎

### 4.3 Empirical bounds via importance sampling

In practice, we only have access to an empirical distribution $\mathbb{P}_n^{\mathbf{V}}$ of sample size $n$ instead of the full observational distribution $\mathbb{P}^{\mathbf{V}}$. We thus obtain estimates $\hat{\mathbb{P}}(w \mid \mathbf{x}, \mathbf{m}_W, \mathbf{a})$ of the conditional density or probability mass functions $\mathbb{P}(w \mid \mathbf{x}, \mathbf{m}_W, \mathbf{a})$ for all $W \in \{M_1, \ldots, M_\ell, Y\}$. If $W$ is discrete, this reduces to a standard (multi-class) classification problem, and the estimated class probabilities can be plugged into Algorithm 1. If $W = Y$ is continuous, we can use arbitrary conditional density estimators to obtain $\hat{\mathbb{P}}(y \mid \mathbf{x}, \bar{\mathbf{m}}_\ell, \mathbf{a})$. We propose an importance sampling approach to estimate $\mathcal{D}\left(\mathbb{P}_+^Y(\cdot \mid \mathbf{x}, \bar{\mathbf{m}}_\ell, \mathbf{a})\right)$ assuming that we can sample from from our estimated density $\hat{\mathbb{P}}^Y(\cdot \mid \mathbf{x}, \bar{\mathbf{m}}_\ell, \mathbf{a})$ (see Appendix F for details and derivations). If $\mathcal{D}$ is the expectation functional, our estimator is

$$\mathcal{D}\left(\widehat{\mathbb{P}_+^Y(\cdot \mid \mathbf{x}, \bar{\mathbf{m}}_\ell, \mathbf{a})}\right) = \frac{1}{\hat{s}_Y^+ k} \sum_{i=1}^{\lfloor kc_Y^+ \rfloor} y_i + \frac{1}{\hat{s}_Y^-} \sum_{i=\lfloor kc_Y^+ \rfloor + 1}^{k} y_i, \quad \text{where} \quad (y_i)_{i=1}^k \sim \hat{\mathbb{P}}^Y(\cdot \mid \mathbf{x}, \bar{\mathbf{m}}_\ell, \mathbf{a}) \quad \text{is sorted.} \tag{8}$$

We also provide importance sampling estimators for distributional effects in Appendix F. We can also obtain empirical confidence intervals by using the same bootstrap procedure as described in [32].

**Implementation:** We use feed-forward neural networks with softmax activation function to estimate discrete probability mass functions. For densities, we use conditional normalizing flows [73] (neural spline flows [21]), which are universal density approximators and allow for sampling. We perform training using the Adam optimizer [41]. We also perform extensive hyperparameter tuning in our experiments. Implementation and hyperparameter tuning details are in Appendix G.

## 5 Experiments

**Baselines:** Most existing methods focus on sensitivity analysis for binary treatments under the MSM [18, 19, 31, 35, 53]. In this setting, our bounds coincide with existing optimality results [18]. For mediation analysis, we are to the best of our knowledge the first to propose bounds for MSM-based sensitivity analysis. This is why we refrain from benchmarking against baselines in our experiments and only show the validity of our bounds.

**Synthetic data:** We perform extensive experiments using synthetic data from various causal inference settings to evaluate the validity of our bounds. Synthetic data are commonly used to evaluate causal inference methods as they ensure that the causal ground truth is available [6, 16, 76]. Here, we generate six different synthetic datasets with $n = 50,000$ samples in the following manner: We first construct three SCMs with binary treatments for the causal graphs in Fig. 1, that is, (i) no mediators, (ii) a single mediator $M$, and (iii) two mediators $M_1$ and $M_2$. We then construct three more SCMs for settings (i)–(iii) with continuous treatments. Details are in Appendix H.

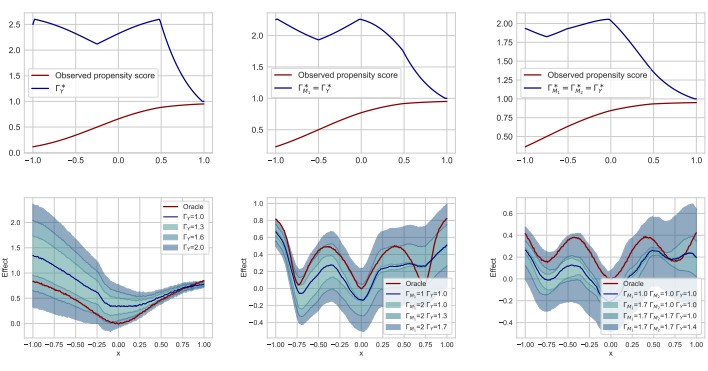

To demonstrate the validity of our bounds, we aim to show that they contain the oracle causal effect whenever the sensitivity constraints are satisfied. We evaluate two versions of our GMSM: the MSM for binary treatments and the CMSM for continuous treatments (see Sec. 3.3). Using oracle knowledge from the SCMs, we estimate the respective density ratio from Eq. (3) and obtain the oracle sensitivity parameter $\Gamma_W^*$ (details are in Appendix H).

Figure 3: Results for the binary treatment setting. Settings (i)–(iii) are ordered from left to right. The top row shows the oracle sensitivity parameter $\Gamma_W^*$ (depending on $x$), and the bottom row shows the bounds.

We thus demonstrate that our bounds contain the oracle causal effect whenever $\Gamma_W \geq \Gamma_W^*$, i.e., whenever we choose sensitivity parameters $\Gamma_W$ at least as large as the oracle $\Gamma_W^*$.

The results for binary treatments are shown in Fig. 3 and for continuous treatments in Fig. 4. For binary treatments, we evaluate the causal effect $Q(\mathbf{x}, \bar{\mathbf{a}}_{\ell+1}, \mathcal{M})$ for over $x \in [-1, 1]$ with three (arbitrary) treatment combinations corresponding to the three different settings: (i) $\bar{\mathbf{a}}_1 = 1$, (ii) $\bar{\mathbf{a}}_2 = (1, 0)$, and (iii) $\bar{\mathbf{a}}_3 = (1, 0, 0)$. For continuous treatments, we also use three (arbitrary) treatment combinations: (i) $\bar{\mathbf{a}}_1 = 0.6$, (ii) $\bar{\mathbf{a}}_2 = (0.9, 0.5)$, and (iii) $\bar{\mathbf{a}}_3 = (0.2, 0.4, 0.5)$. Results for additional treatment combinations are in Appendix J. Evidently, the oracle causal effect is contained within our bounds. Hence, the results confirm the validity of our bounds.

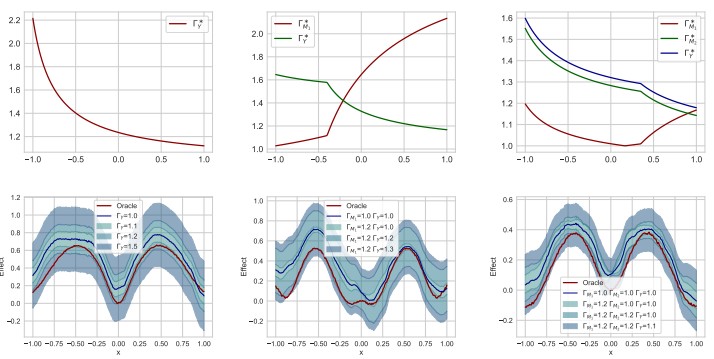

Figure 4: Results for continuous treatment setting. Settings (i)–(iii) are ordered from left to right. The top row shows the oracle sensitivity parameter $\Gamma_W^*$ (depending on $x$), and the bottom row shows the bounds.

We also compare the bounds from Jesson et al. [32] with our sharp bounds under the CMSM and examine whether we can improve on these by using a different weight function (Def. 4). For this purpose, we modify the SCM from setting (i) in the continuous treatment setting so that there is no unobserved confounding for individuals with $x > 0$ (i.e., $\Gamma_Y^* = 0$). Details are in Appendix H. Table 1 reports the bounds from Jesson et al., our bounds under the CMSM, and our bounds under a weighted CMSM with weight function $q_Y(x) = \mathbb{1}(x > 0)$. The two main findings are: (i) Using the weight function, we can leverage prior knowledge about the confounding structure to obtain even tighter bounds. (ii) Our bounds are much faster to compute (using $5{,}000$ samples). The latter is because we derived closed-form solutions for our bounds, while the method proposed by Jesson et al. is an approximation that uses grid search.

Table 1: Experiment with weighted CMSM

| Bounds $\diagdown$ Metric/ $\Gamma_Y$ | Interval length | | | Coverage | | | Time (sec.) |
|---|---|---|---|---|---|---|---|
| | 1.2 | 1.5 | 2 | 1.2 | 1.5 | 2 | |
| Jesson et al.[32] (CMSM) | $0.33 \pm 0.00$ | $0.74 \pm 0.01$ | $1.27 \pm 0.01$ | 1 | 1 | 1 | $137.48 \pm 2.02$ |
| Our bounds (CMSM) | $0.33 \pm 0.00$ | $0.74 \pm 0.01$ | $1.25 \pm 0.01$ | 1 | 1 | 1 | $0.39 \pm 0.02$ |
| Our bounds (weighted CMSM) | $\mathbf{0.17 \pm 0.01}$ | $\mathbf{0.37 \pm 0.02}$ | $\mathbf{0.63 \pm 0.03}$ | 0.6 | 1 | 1 | $0.42 \pm 0.05$ |

Reported: Average $\pm$ standard deviation over 5 random seeds (best in bold).

**Real-world data:** We demonstrate our bounds using an example with real-world data. We consider a setting from the COVID-19 pandemic where mobility (captured through telephone movement) was monitored to obtain a leading predictor of case growth [57]. Details regarding the data (publicly available) and our analysis are in Appendix I. Here, mobility is the mediator, case growth is the outcome, and stay-home order (ban of gatherings with more than 5 people) is the treatment. We are interested in the natural directed effect (NDE, see Example 2) of a stay-home order on the case growth. We suspect that unobserved confounders between treatment and mediator might exist (e.g., adherence, etc. of the population). Hence we perform a causal sensitivity analysis where we vary $\Gamma_M$ and plot the bounds in Fig. 5. We observe that the estimated effect under unconfoundedness ($\Gamma_M = 1$) is negative, i.e., the stay-home order decreases case growth. For $\Gamma_M > 1$, we obtain bounds around this estimand. For $\Gamma_M < 6$ the bounds are negative, which means that under moderate unobserved confounding, it seems likely that the NDE is nonzero, in line with prior evidence [57].

## 6 Discussion

**Assumptions:** As common in the causal inference literature, our results rely on assumptions on the data-generating process that must be justified by domain knowledge. Our main assumption is that we exclude confounders between mediators and the outcome in our analysis (see Def. 2). The reason

for this assumption is that it allows us to interpret the causal query from Eq. (1) as a path-specific effect. Path-specific effects are defined as so-called nested counterfactuals that lie in the third layer of Pearl's causal hierarchy [55] Using the assumption from Theorem 1, they can be reduced to the query from Eq. (1), which lies in layer 2 of Pearl's hierarchy, i.e., only depends on interventions [15]. Our sensitivity analysis then bridges the gap from layer 2 to layer 1 (observational data). Approaches for relaxing this assumption, e.g., by combining our results with a sensitivity analysis from layer 3 to layer 2, are left for future work.

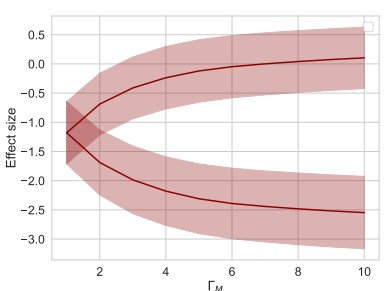

Figure 5: Estimated upper/lower bound for the NDE on real-world data. Reported: mean and standard deviation over 10 runs.

**Sensitivity parameter and weighting function:** Both the sensitivity parameter $\Gamma_W$ and the weighting function $q(\mathbf{a}, \mathbf{x})$ incorporate domain knowledge about the unobserved confounding and need to be chosen by the practitioner accordingly.

The sensitivity parameter $\Gamma_W$ controls the strength of unobserved confounding. In practice, one typically chooses $\Gamma$ by domain knowledge or data-driven heuristics [35, 28]. One approach is to obtain the smallest $\Gamma$ so that the corresponding partially identified interval includes 0. Then, $\Gamma$ can be interpreted as a level of "causal uncertainty", quantifying the smallest violation of unconfoundedness that would explain away the causal effect [31, 33].

The weight function $q(\mathbf{a}, \mathbf{x})$ offers additional opportunities to incorporate domain knowledge about the confounding structure to obtain tighter bounds. Consider an observational study on the effect of smoking on cancer risk, confounded by certain unobserved genes. For example, it may be known that genes that act as unobserved confounders do not affect the cancer risk for a certain population with certain covariates $\mathbf{x}$, which allows us to set $q(\mathbf{a}, \mathbf{x}) = 1$. Note that we can always set $q(\mathbf{a}, \mathbf{x}) = \mathbb{P}(\mathbf{a} \mid \mathbf{x})$ (discrete treatments) or $q(\mathbf{a}, \mathbf{x}) = 0$ (continuous treatments) if no domain knowledge is available, leading to established sensitivity models from the literature. For example, practitioners may use our bounds for the MSM in a mediation analysis setting without ever explicitly using the GMSM formulation via a weighting function.

**Other sensitivity models:** In this paper, we provide bounds for MSM-type sensitivity models. Recently, other types of sensitivity models have been proposed in the literature, which may provide less conservative bounds in situations where the data-generating process does not follow an MSM. Examples include $f$-sensitivity models [34], $L_2$-sensitivity models [81], curvature sensitivity models [48] and the $\delta$-MSM [45]. Extending our results to these sensitivity models may be another possible direction for future work.

**Efficient estimation:** Our main results (Theorem 1, Corollary 1, 2) are *identifiability results*, i.e., hold in the limit of infinite data. We did not provide results on efficient estimation. Therefore, future work may consider extending our approach to incorporate semiparametric efficiency theory [37].

**Conclusion:** We proposed a flexible generalization of the MSM and derived sharp bounds for a variety of different causal inference settings and data types (e.g., continuous, categorical). Our work provides practitioners with a unified framework for causal sensitivity analysis. This enables reliable causal inferences from observational data in the presence of unobserved confounders, thus promoting safe decision-making.

## Acknowledgments

SF acknowledges funding from the Swiss National Science Foundation (SNSF) via Grant 186932.

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

# A Extended related work

## A.1 Partial identification under unobserved confounding

There are various works for partial identification (i.e., bounding causal effects) under unobserved confounding that do not impose sensitivity models. In the following, we discuss the difference between this literature stream, which we call causal partial identification (CPA), to causal sensitivity analysis (CSA).

The main difference between CSA and CPA is that CSA imposes sensitivity models, that is, assumptions on the strength of unobserved confounding, which is controlled by a sensitivity parameter $\Gamma$. Methods for CPA do not impose sensitivity constraints but instead impose other assumptions. In practice, CSA can be used to test the robustness of causal effect estimates to violations of the unconfoundedness assumption (by varying $\Gamma$), while CPA may be applicable in situations where no domain knowledge about the confounding strength is available.

Approaches for CPA can be roughly described by two categories that describe the type of assumptions that are imposed to derive informative bounds.

1. **Additional variables:** In order to achieve informative bounds without restricting the strength of unobserved confounding, some works impose assumptions on the data-generating process by postulating the existence of additional variables. One example is instrumental variables (IVs), i.e., variables, which only have a direct effect on treatment variables but not on outcomes. Under certain assumptions, IVs render bounds for causal effects informative without assumptions on the underlying confounding structure [3, 25, 40, 54]. Other examples include leaky mediation [3, 54], differential effects [11], noisy proxy settings [26].

2. **Discrete SCMs:** Another stream of literature derives informative bounds in *discrete* SCMs. Examples include [20, 75, 79, 80].

**Comparison with our bounds**: None of the methods above is applicable in the causal inference settings we consider (e.g., continuous treatments or outcomes, no IVs available, etc.). In contrast, a well-known CPA result in our setting is the so-called no-assumptions bound [44] that leverages discrete treatments. In the following, we consider the standard setting for CATE estimation without mediators ($\ell = 0$) and a binary treatment $A \in \{0, 1\}$. We show that we obtain the no-assumptions bound with our sensitivity analysis for the query $\mathbb{E}[Y \mid x, do(A = a)]$ for $\Gamma \to \infty$, i.e. when lifting the sensitivity constraint. Let $[p_1, p_2]$ denote the support of the conditional density $\mathbb{P}(y \mid x, a)$ with c.d.f. $F$. Using the MSM we obtain

$$Q^+(\mathbf{x}, a, \mathcal{S}) = \int_{p_1}^{F^{-1}(\frac{\Gamma}{1+\Gamma})} y\left((1 - \Gamma^{-1})\mathbb{P}(a \mid \mathbf{x}) + \Gamma^{-1}\right) \mathbb{P}(y \mid \mathbf{x}, a)\, dy \tag{9}$$

$$+ \int_{F^{-1}(\frac{\Gamma}{1+\Gamma})}^{p_2} y\left((1 - \Gamma)\mathbb{P}(a \mid \mathbf{x}) + \Gamma\right) \mathbb{P}(y \mid \mathbf{x}, a)\, dy \tag{10}$$

$$\xrightarrow[\Gamma \to \infty]{} \mathbb{P}(a \mid \mathbf{x})\, \mathbb{E}[Y \mid \mathbf{x}, a] + (1 - \mathbb{P}(a \mid \mathbf{x}))\, p_2, \tag{11}$$

which corresponds to the result in [44]. The result for the lower bound $Q^-(\mathbf{x}, a, \mathcal{S})$ follows analogously.

## A.2 Estimation of causal effects under unconfoundedness

Under certain additional assumptions, unconfoundedness makes it possible to point-identify causal effects from the observational data, so that the causal inference problem reduces to a purely statistical estimation problem. Various methods for estimating point-identified causal effects under unconfoundedness have been proposed that make use of machine learning and/or (semiparametric) statistical theory. Examples include methods for conditional average treatment effects [13, 16, 38, 42, 63, 72, 78], average treatment effects [24, 64, 69], instrumental variables [5, 23, 27, 65, 66, 76], time-varying data [8, 43, 46], mediation analysis [68, 22], and distributional effects [12, 39, 47, 49]. Note that all the methods above are biased if the unconfoundedness assumption is violated, which outlines the need for our causal sensitivity analysis.

## B Proofs of the GMSM bounds

### B.1 Proof of Theorem 1

*Proof.* We give the proof for continuous $W \in \mathbb{R}$ (see Eq. (4) in the main paper). The derivation for discrete $W \in \mathbb{N}$ (see Eq. (5) in the main paper) follows with the same arguments and the normalization constraint $\sum_w \mathbb{P}^+(w \mid \mathbf{x}, \mathbf{m}_W, \mathbf{a}) = 1$. We first show the inequality

$$F_+(w) \leq \inf_{\mathcal{M} \in \mathcal{C}(\mathcal{S})} F_{\mathcal{M}}(w) \tag{12}$$

and then provide sharpness results by showing

$$F_+(w) \geq \inf_{\mathcal{M} \in \mathcal{C}(\mathcal{S})} F_{\mathcal{M}}(w) \tag{13}$$

under the sharpness conditions of Theorem 1. The result for $F_-(w)$ follows analogously.

**Validity of the bounds ($\leq$):** We provide a proof by contradiction. To do so, we assume that there exists an SCM $\mathcal{M} \in \mathcal{C}(\mathcal{S})$ and $w \in \mathbb{R}$ so that

$$F_+(w) > F_{\mathcal{M}}(w). \tag{14}$$

By the definition of $F_+(w)$, there must exist a set $\mathcal{W}_1 \subseteq \mathbb{R}_{\leq F^{-1}(c_W^+)}$, so that

$$\mathbb{P}_+(w_1 \mid \mathbf{x}, \mathbf{m}_W, \mathbf{a}) > \mathbb{P}(w_1 \mid \mathbf{x}, \mathbf{m}_W, do(\mathbf{A} = \mathbf{a})) \quad \text{for all} \quad w_1 \in \mathcal{W}_1, \tag{15}$$

or a set $\mathcal{W}_2 \subseteq \mathbb{R}_{> F^{-1}(c_W^+)}$, so that

$$\mathbb{P}_+(w_2 \mid \mathbf{x}, \mathbf{m}_W, \mathbf{a}) < \mathbb{P}(w_2 \mid \mathbf{x}, \mathbf{m}_W, do(\mathbf{A} = \mathbf{a})) \quad \text{for all} \quad w_2 \in \mathcal{W}_2, \tag{16}$$

as otherwise $\mathbb{P}(w \mid \mathbf{x}, \mathbf{m}_W, do(\mathbf{A} = \mathbf{a}))$ would not integrate to 1. Let $W = f(\mathbf{X}, \mathbf{M}_W, \mathbf{A}, \mathbf{U}_W)$ be the functional assignment of $\mathcal{M}$ and let $\mathcal{U}_1 = f_{\mathbf{x},\mathbf{m}_W,\mathbf{a}}^{-1}(\mathcal{W}_1) \subseteq \mathbb{R}^d$ and $\mathcal{U}_2 = f_{\mathbf{x},\mathbf{m}_W,\mathbf{a}}^{-1}(\mathcal{W}_2) \subseteq \mathbb{R}^d$ denote the preimages of $\mathcal{W}_1$ and $\mathcal{W}_2$ under $f_{\mathbf{x},\mathbf{m}_W,\mathbf{a}}$ in the confounding space.

We can again write $\mathbb{P}_+(w \mid \mathbf{x}, \mathbf{m}_W, \mathbf{a})$ as a push forward

$$\mathbb{P}^+(w \mid \mathbf{x}, \mathbf{m}_W, \mathbf{a}) = f_{\mathbf{x},\mathbf{m}_W,\mathbf{a}\#} \mathbb{P}_+^{\mathbf{U}_W|\mathbf{x},\mathbf{a}}(w) \tag{17}$$

for some density $\mathbb{P}_+(\mathbf{u}_W \mid \mathbf{x}, \mathbf{a})$ on the confounding space. By the definition of $\mathbb{P}_+(w \mid \mathbf{x}, \mathbf{m}_W, \mathbf{a})$ and Eq. (28), we obtain

$$\mathbb{P}_+(\mathbf{u}_1 \mid \mathbf{x}, \mathbf{a}) = \frac{1}{s_W^+} \mathbb{P}(\mathbf{u}_1 \mid \mathbf{x}, \mathbf{a}) \quad \text{and} \quad \mathbb{P}_+(\mathbf{u}_2 \mid \mathbf{x}, \mathbf{a}) = \frac{1}{s_W^-} \mathbb{P}(\mathbf{u}_2 \mid \mathbf{x}, \mathbf{a}) \tag{18}$$

for all $\mathbf{u}_1 \in \mathcal{U}_1$ and $\mathbf{u}_2 \in \mathcal{U}_2$. Due to the definition of $\mathcal{U}_1$ and $\mathcal{U}_2$, it follows that there exist $\mathbf{u}_1 \in \mathcal{U}_1$ and $\mathbf{u}_2 \in \mathcal{U}_2$, so that

$$\frac{\mathbb{P}(\mathbf{u}_1 \mid \mathbf{x}, \mathbf{a})}{\mathbb{P}(\mathbf{u}_1 \mid \mathbf{x}, do(\mathbf{A} = \mathbf{a}))} > \frac{\mathbb{P}(\mathbf{u}_1 \mid \mathbf{x}, \mathbf{a})}{\mathbb{P}_+(\mathbf{u}_1 \mid \mathbf{x}, \mathbf{a})} = s_W^+ \tag{19}$$

and

$$\frac{\mathbb{P}(\mathbf{u}_1 \mid \mathbf{x}, \mathbf{a})}{\mathbb{P}(\mathbf{u}_1 \mid \mathbf{x}, do(\mathbf{A} = \mathbf{a}))} < \frac{\mathbb{P}(\mathbf{u}_1 \mid \mathbf{x}, \mathbf{a})}{\mathbb{P}_+(\mathbf{u}_1 \mid \mathbf{x}, \mathbf{a})} = s_W^-. \tag{20}$$

Both Eq. (19) and Eq. (20) are contradictions to the GMSM constraint Eq. (3) of the main paper. Hence, $\mathcal{M} \notin \mathcal{C}(\mathcal{S})$.

**Sharpness of the bounds ($\geq$):** We show that under the sharpness conditions from Theorem 1, there exists an SCM $\mathcal{M} \in \mathcal{C}(\mathcal{S})$ with induced interventional density $\mathbb{P}_+(w \mid \mathbf{x}, \mathbf{m}_W, \mathbf{a})$ for all $w$. The construction of $\mathcal{M}$ is similar to that of our motivational toy example in Sec. 4.1 of the main paper. We first define an (interventional) probability density for the unobserved confounder $\mathbf{U}_W \in \mathbb{R}^d$ given $\mathbf{X}$ via

$$\mathbb{P}(\mathbf{u}_W \mid \mathbf{x}, do(\mathbf{A} = \mathbf{a})) = \mathbb{1}(0 \leq u_W^{(1)} \leq c_W^+)(1/s_W^+) + \mathbb{1}(1 \geq u_W^{(1)} > c_W^+)(1/s_W^-), \tag{21}$$

where $u_W^{(1)}$ denotes the first coordinate of $\mathbf{u}_W$. $\mathbb{P}(\mathbf{u}_W \mid \mathbf{x}, \mathbf{m}_W)$ is a properly normalized density with support $[0, 1]^d$ because

$$\int \mathbb{P}(\mathbf{u}_W \mid \mathbf{x}, do(\mathbf{A} = \mathbf{a})) \, d\mathbf{u}_W = \frac{c_W^+}{s_W^+} + \frac{1 - c_W^+}{s_W^-} = \frac{1 - s_W^-}{s_W^+ - s_W^-} + \frac{s_W^+ - 1}{s_W^+ - s_W^-} = 1, \tag{22}$$

where we used the definition of $c_W^+ = \frac{(1-s_W^-)s_W^+}{s_W^+ - s_W^-}$.

We now define the probability density for the unobserved confounder $\mathbf{U}_W \in \mathbb{R}^d$ given $\mathbf{X}$, $\mathbf{M}_W$, and treatments $\mathbf{A}$ as the uniform density on $[0,1]^d$, i.e.,

$$\mathbb{P}(\mathbf{u}_W \mid \mathbf{x}, \mathbf{a}) = \mathbb{1}(0 \leq \mathbf{u}_W \leq 1), \tag{23}$$

where the indicator function is defined coordinate-wise.

We now need to show that there always exists an SCM $\mathcal{M}$ which induces the densities in Eq. (21) and Eq. (22) and that satisfies the sensitivity constraints. For this purpose, we use the assumptions to write the interventional density for discrete $\mathbf{A}$ as

$$\mathbb{P}(\mathbf{u}_W \mid \mathbf{x}, do(\mathbf{A} = \mathbf{a}))) = \mathbb{P}(\mathbf{u}_W \mid \mathbf{x}) = \mathbb{P}(\mathbf{a} \mid \mathbf{x}) + (1 - \mathbb{P}(\mathbf{a} \mid \mathbf{x}))\mathbb{P}(\mathbf{u}_W \mid \mathbf{x}, \mathbf{A} \neq \mathbf{a}). \tag{24}$$

Hence, our definitions for $\mathbb{P}(\mathbf{u}_W \mid \mathbf{x}, do(\mathbf{A}))$ and $\mathbb{P}(\mathbf{u}_W \mid \mathbf{x}, \mathbf{a})$ are valid whenever

$$0 \leq \mathbb{P}(\mathbf{u}_W \mid \mathbf{x}, \mathbf{A} \neq \mathbf{a}) = \frac{\mathbb{P}(\mathbf{u}_W \mid \mathbf{x}, do(\mathbf{A} = \mathbf{a})) - \mathbb{P}(\mathbf{a} \mid \mathbf{x})}{1 - \mathbb{P}(\mathbf{a} \mid \mathbf{x})}, \tag{25}$$

which follows from the sharpness condition $1/s_W^+ \geq \mathbb{P}(\mathbf{a} \mid \mathbf{x})$. Sharpness for continuous $\mathbf{A}$ follows because we can approximate $\mathbf{A}$ with a sequence of discrete treatments $(\mathbf{A}_n)_n$ and it holds that $\mathbb{P}(\mathbf{a}_n \mid \mathbf{x}) \xrightarrow[n \to \infty]{} 0$ (as long as $\mathbb{P}(\mathbf{a} \mid \mathbf{x})$ is non-degenerate). Finally, it holds that

$$s_W^- \leq \frac{\mathbb{P}(\mathbf{U}_W = \mathbf{u}_W \mid \mathbf{x}, \mathbf{a})}{\mathbb{P}(\mathbf{U}_W = \mathbf{u}_W \mid \mathbf{x}, do(\mathbf{A} = \mathbf{a}))} \leq s_W^+ \quad \text{for all} \quad \mathbf{u}_W \in [0,1]^p, \tag{26}$$

so that $\mathcal{M}$ respects the sensitivity constraint of the GMSM $\mathcal{S}$.

Let now $\mathbb{P}(w \mid \mathbf{x}, \mathbf{m}_W, \mathbf{a})$ denote the observational density of $W$ given $\mathbf{X}$, $\mathbf{M}_W$, and $\mathbf{A}$ with corresponding cumulative distribution function (CDF) given by $F(w)$. To complete our construction of $\mathcal{M}$, we define functional assignment $W = f_W(\mathbf{X}, \mathbf{M}_W, \mathbf{A}, \mathbf{U}_W)$ via the inverse CDF

$$f_W(\mathbf{x}, \mathbf{m}_W, \mathbf{a}, \mathbf{u}_W) = F^{-1}\left(u_W^{(1)}\right). \tag{27}$$

By denoting $f_W(\mathbf{x}, \mathbf{m}_W, \mathbf{a}, \cdot)$ as $f_{\mathbf{x}, \mathbf{m}_W, \mathbf{a}}$, we can write the observational distribution under $\mathcal{M}$ as the push forward

$$f_{\mathbf{x}, \mathbf{m}_W, \mathbf{a}\#} \mathbb{P}^{\mathbf{U}_W \mid \mathbf{x}, \mathbf{a}}(w) = \mathbb{P}(w \mid \mathbf{x}, \mathbf{m}_W, \mathbf{a}) \tag{28}$$

due to Eq. (22) and Eq. (27). Note that we used here that $\mathbf{U}_W$ is not a parent of $\mathbf{M}_W$ (see Fig. 1). Hence, $\mathcal{M} \in \mathcal{C}(\mathcal{S})$ is compatible with the sensitivity model $\mathcal{S}$. Furthermore, the induced interventional distribution can be written as the push-forward

$$f_{\mathbf{x}, \mathbf{m}_W, \mathbf{a}\#} \mathbb{P}^{\mathbf{U}_W \mid \mathbf{x}, do(\mathbf{A}=\mathbf{a})}(w) = \mathbb{P}_+(w \mid \mathbf{x}, \mathbf{m}_W, \mathbf{a}) \tag{29}$$

because of Eq. (21). □

## B.2 Proof of Corollary 1

Here, we formally restate Corollary 1 for *monotone* functionals. For two probability densities $\mathbb{P}(y)$ and $\mathbb{P}'(y)$, we denote $\mathbb{P} \leq \mathbb{P}'$ if $F \geq F'$ holds almost surely for the corresponding CDFs.

**Definition 5.** A functional $\mathcal{D}$ is called monotone if $\mathcal{D}(\mathbb{P}(\cdot)) \leq \mathcal{D}(\mathbb{P}'(\cdot))$ whenever $\mathbb{P} \leq \mathbb{P}'$.

Intuitively, a monotone functional increases if applied on a distribution that is further right-shifted. Note that both the expectation functional $\mathcal{D}(\mathbb{P}(\cdot)) = \int y\mathbb{P}(y)\,dy$ and the quantile functionals $\mathcal{D}(\mathbb{P}(\cdot)) = F^{-1}(\alpha)$ for $\alpha \in [0,1]$ are monotone.

**Corollary 3** (Restatement)**.** *If $\mathbf{M} = \emptyset$ and $\mathcal{D}$ is monotone, we obtain sharp bounds*

$$Q^+(\mathbf{x}, \mathbf{a}, \mathcal{S}) = \mathcal{D}\left(\mathbb{P}_+^Y(\cdot \mid \mathbf{x}, \mathbf{a})\right) \quad \text{and} \quad Q^-(\mathbf{x}, \mathbf{a}, \mathcal{S}) = \mathcal{D}\left(\mathbb{P}_-^Y(\cdot \mid \mathbf{x}, \mathbf{a})\right). \tag{30}$$

*Proof.* Follows directly from Theorem 1 for $W = Y$. □

## B.3 Proof of Corollary 2

*Proof.* We derive Algorithm 1 for $Q^+(\mathbf{x}, \bar{\mathbf{a}}_{\ell+1}, \mathcal{S})$. The case for $Q^-(\mathbf{x}, \bar{\mathbf{a}}_{\ell+1}, \mathcal{S})$ follows analogously.

Recall that we want to maximize the causal effect

$$Q(\mathbf{x}, \bar{\mathbf{a}}, \mathcal{M}) = \sum_{\mathbf{m}} \mathcal{D}\left(\mathbb{P}^Y(\cdot \mid \mathbf{x}, \mathbf{m}, do(\mathbf{A} = \mathbf{a}_{\ell+1}))\right) \prod_{i=1}^{\ell} \mathbb{P}(m_i \mid \mathbf{x}, \bar{\mathbf{m}}_{i-1}, do(\mathbf{A} = \mathbf{a}_i)), \quad (31)$$

over all possible SCMs $\mathcal{M} \in \mathcal{C}(\mathcal{S})$ that are compatible with the GMSM $\mathcal{S}$. By using the assumption (no unobserved confounding between mediators and outcome), we can write $Q(\mathbf{x}, \bar{\mathbf{a}}, \mathcal{M})$ in terms of functional assignments $f_{\mathbf{x}, \mathbf{m}_W, \mathbf{a}}^W$ defined via $W = f^W(\mathbf{X}, \mathbf{M}_W, \mathbf{A}, \mathbf{U}_W)$ and induced (interventional) distributions $\mathbb{P}^{\mathbf{U}_W | \mathbf{x}}$ in the following way:

$$Q(\mathbf{x}, \bar{\mathbf{a}}_{\ell+1}, \mathcal{M}) = \sum_{\mathbf{m}} \mathcal{D}\left(f_{\mathbf{x}, \bar{\mathbf{m}}_\ell, \mathbf{a}_{\ell+1} \#}^Y \mathbb{P}^{\mathbf{U}_Y | \mathbf{x}}(\cdot)\right) \prod_{i=1}^{\ell} f_{\mathbf{x}, \bar{\mathbf{m}}_{i-1}, \mathbf{a}_i \#}^{M_i} \mathbb{P}^{\mathbf{U}_{M_i} | \mathbf{x}}(m_i). \quad (32)$$

Hence, the optimization problem reduces to maximizing Eq. (32) over all functional assignments $f_{\mathbf{x}, \mathbf{m}_W, \mathbf{a}}^W$ and distributions $\mathbb{P}^{\mathbf{U}_W | \mathbf{x}}$ that are compatible with $\mathcal{S}$. Note that the terms in the product do not depend on each other or the term in the sum. Thus, by rearranging the suprema and products, we can equivalently perform the following iterative procedure: First, we initialize

$$Q_{\ell+1}^+(\mathbf{x}, \bar{\mathbf{m}}_\ell, \bar{\mathbf{a}}_{\ell+1}, \mathcal{S}) = \sup_{\mathcal{M} \in \mathcal{C}(\mathcal{S})} \mathcal{D}\left(f_{\mathbf{x}, \bar{\mathbf{m}}_\ell, \mathbf{a}_{\ell+1} \#}^Y \mathbb{P}^{\mathbf{U}_Y | \mathbf{x}}(\cdot)\right) \quad (33)$$

and then define

$$Q_i^+(\mathbf{x}, \bar{\mathbf{m}}_{i-1}, \bar{\mathbf{a}}_{\ell+1}, \mathcal{S}) = \sup_{\mathcal{M} \in \mathcal{C}(\mathcal{S})} \sum_{m_i} Q_{i+1}^+(\mathbf{x}, \bar{\mathbf{m}}_{i-1}, m_i, \bar{\mathbf{a}}_{\ell+1})) \left(f_{\mathbf{x}, \bar{\mathbf{m}}_{i-1}, \mathbf{a}_i \#}^{M_i} \mathbb{P}^{\mathbf{U}_Y | \mathbf{x}}(m_i)\right) \quad (34)$$

for all $i \in \{\ell, \ldots, 1\}$, which results in the sharp upper bound

$$Q^+(\mathbf{x}, \bar{\mathbf{a}}_{\ell+1}, \mathcal{S}) = Q_1^+(\mathbf{x}, \bar{\mathbf{a}}_{\ell+1}, \mathcal{S}). \quad (35)$$

For Eq. (33) and monotone $\mathcal{D}$, we can directly apply Theorem 1 and obtain

$$Q_{\ell+1}^+(\mathbf{x}, \bar{\mathbf{m}}_\ell, \bar{\mathbf{a}}_{\ell+1}, \mathcal{S}) = \mathcal{D}\left(\mathbb{P}_+^Y(\cdot \mid \mathbf{x}, \bar{\mathbf{m}}_\ell, \mathbf{a}_{\ell+1})\right). \quad (36)$$

For Eq. (34), we need to find an induced distribution $f_{\mathbf{x}, \bar{\mathbf{m}}_{i-1}, \mathbf{a}_i \#}^{M_i} \mathbb{P}^{\mathbf{U}_Y | \mathbf{x}}$ on $M_i$ that is compatible with $\mathcal{S}$ and puts most probability mass on $m_i$ where $Q_{i+1}^+(\mathbf{x}, \bar{\mathbf{m}}_{i-1}, m_i, \bar{\mathbf{a}}_{\ell+1}))$ is large. Hence, we can apply the discrete version of Theorem 1 with $W = \pi(M_i)$, where $\pi \colon supp(M_i) \to supp(M_i)$ is a permutation map so that $\left(Q_{i+1}^+(\mathbf{x}, \bar{\mathbf{m}}_{i-1}, \pi(m_i)), \bar{\mathbf{a}}_{\ell+1}\right)_{m_i \in supp(M_i)}$ is ordered in ascending order. The corresponding update step is shown in Algorithm 1.

$\square$

## C  Special cases of the GMSM

In this section, we prove Lemma 1, i.e., we show that all sensitivity models introduced in Sec. 3.3 of the main paper are special cases of our (weighted) GMSM. Recall that we consider settings without mediators (i.e., $\ell = 0$), and write $\Gamma = \Gamma_Y$ for the sensitivity parameter, $q(\mathbf{a}, \mathbf{x}) = q_Y(\mathbf{a}, \mathbf{x})$ for the weight function, and $\mathbf{U} = \mathbf{U}_Y$ for the unobserved confounders. In this case, the weighted GMSM is defined via the confounding restriction

$$\frac{1}{(1 - \Gamma)\, q(\mathbf{a}, \mathbf{x}) + \Gamma} \leq \frac{\mathbb{P}(\mathbf{U} = \mathbf{u} \mid \mathbf{x}, \mathbf{a})}{\mathbb{P}(\mathbf{U} = \mathbf{u} \mid \mathbf{x}, do(\mathbf{A} = \mathbf{a}))} \leq \frac{1}{(1 - \Gamma^{-1})\, q(\mathbf{a}, \mathbf{x}) + \Gamma^{-1}}. \tag{37}$$

### C.1  Marginal sensitivity model (MSM):

The MSM [67] for binary treatment $\mathbf{A} = A \in \{0, 1\}$ is defined via

$$\frac{1}{\Gamma} \leq \frac{\pi(\mathbf{x})}{1 - \pi(\mathbf{x})} \frac{1 - \pi(\mathbf{x}, \mathbf{u})}{\pi(\mathbf{x}, \mathbf{u})} \leq \Gamma, \tag{38}$$

where $\pi(\mathbf{x}) = \mathbb{P}(A = 1 \mid \mathbf{x})$ denotes the observed propensity score and $\pi(\mathbf{x}, \mathbf{u}) = \mathbb{P}(A = 1 \mid \mathbf{x}, \mathbf{u})$ denotes the full propensity score. By rearranging the terms, we obtain

$$\frac{1}{(1 - \Gamma)\mathbb{P}(a \mid \mathbf{x}) + \Gamma} \leq \frac{\mathbb{P}(a \mid \mathbf{x}, \mathbf{u})}{\mathbb{P}(a \mid \mathbf{x})} \leq \frac{1}{(1 - \Gamma^{-1})\mathbb{P}(a \mid \mathbf{x}) + \Gamma^{-1}} \tag{39}$$

for $a \in \{0, 1\}$. Furthermore, by Bayes' theorem, it follows that

$$\frac{\mathbb{P}(a \mid \mathbf{x}, \mathbf{u})}{\mathbb{P}(a \mid \mathbf{x})} = \frac{\mathbb{P}(\mathbf{u} \mid \mathbf{x}, a)\mathbb{P}(a \mid \mathbf{x})}{\mathbb{P}(\mathbf{u} \mid \mathbf{x})\mathbb{P}(a \mid \mathbf{x})} = \frac{\mathbb{P}(\mathbf{u} \mid \mathbf{x}, a)}{\mathbb{P}(\mathbf{u} \mid \mathbf{x})} = \frac{\mathbb{P}(\mathbf{u} \mid \mathbf{x}, a)}{\mathbb{P}(\mathbf{u} \mid \mathbf{x}, do(A = a))}, \tag{40}$$

which implies that the MSM is a weighted GMSM with weight function $q(\mathbf{a}, \mathbf{x}) = \mathbb{P}(\mathbf{a} \mid \mathbf{x})$.

**Comparison to the MSM definition using potential outcomes:** Some papers define the MSM in terms of potential outcomes $(Y_1, Y_0)$ instead of unobserved confounders $U$ [31, 67]. Here, $Y_a$ denotes the potential outcome under the treatment intervention $do(A = a)$. In the following, we show that this is equivalent to the definition we use in our paper.

**Lemma 2.** *The following two statements are equivalent:*

1. *There exists an unobserved confounder $U$ that satisfies $Y_1, Y_0 \perp A \mid \mathbf{X}, \mathbf{U}$ so that it holds that $s^- \leq \frac{\mathbb{P}(\mathbf{u}\mid\mathbf{x},a)}{\mathbb{P}(\mathbf{u}\mid\mathbf{x})} \leq s^+$.*

2. *It holds that $s^- \leq \frac{\mathbb{P}(Y_1,Y_0\mid\mathbf{x},a)}{\mathbb{P}(Y_1,Y_0\mid\mathbf{x})} \leq s^+$.*

*Proof.* The second statement follows directly from the first one by defining $\mathbf{U} = (Y_1, Y_0)$. For the other direction, we proceed via proof by contradiction. Assume there exists a pair $(Y_1, Y_0)$ that violates the second statement, say w.l.o.g. $\frac{\mathbb{P}(Y_1,Y_0\mid\mathbf{x},a)}{\mathbb{P}(Y_1,Y_0\mid\mathbf{x})} > s^+$. We can use the independence condition from the first statement to write

$$\mathbb{P}(Y_1, Y_0 \mid \mathbf{x}, a) = \int \mathbb{P}(Y_1, Y_0 \mid \mathbf{x}, \mathbf{u}, a)\mathbb{P}(\mathbf{u} \mid \mathbf{x}, a)\, \mathrm{d}\mathbf{u} = \int \mathbb{P}(Y_1, Y_0 \mid \mathbf{x}, \mathbf{u})\mathbb{P}(\mathbf{u} \mid \mathbf{x}, a)\, \mathrm{d}\mathbf{u} \tag{41}$$

Furthermore, we have that

$$\mathbb{P}(Y_1, Y_0 \mid \mathbf{x}) = \int \mathbb{P}(y_1, y_0 \mid \mathbf{x}, \mathbf{u})\mathbb{P}(\mathbf{u} \mid \mathbf{x})\, \mathrm{d}\mathbf{u}. \tag{42}$$

It follows that

$$s^+ < \frac{\int \mathbb{P}(Y_1, Y_0 \mid \mathbf{x}, \mathbf{u})\mathbb{P}(\mathbf{u} \mid \mathbf{x}, a)\, \mathrm{d}\mathbf{u}}{\int \mathbb{P}(Y_1, Y_0 \mid \mathbf{x}, \mathbf{u})\mathbb{P}(\mathbf{u} \mid \mathbf{x})\, \mathrm{d}\mathbf{u}}. \tag{43}$$

Hence, there exists a $\mathbf{u}$ such that $s^+ < \frac{\mathbb{P}(Y_1,Y_0\mid\mathbf{x},\mathbf{u})\mathbb{P}(\mathbf{u}\mid\mathbf{x},a)}{\mathbb{P}(Y_1,Y_0\mid\mathbf{x},\mathbf{u})\mathbb{P}(\mathbf{u}\mid\mathbf{x})} = \frac{\mathbb{P}(\mathbf{u}\mid\mathbf{x},a)}{\mathbb{P}(\mathbf{u}\mid\mathbf{x})}$, which is a contradiction to the sensitivity constraint of the first statement. $\square$

### C.2 Continuous marginal sensitivity model (CMSM):

For continuous treatments $\mathbf{A} \in \mathbb{R}^d$, the continuous marginal sensitivity model (CMSM) [32] is defined via

$$\frac{1}{\Gamma} \leq \frac{\mathbb{P}(\mathbf{a} \mid \mathbf{x}, \mathbf{u})}{\mathbb{P}(\mathbf{a} \mid \mathbf{x})} \leq \Gamma. \tag{44}$$

With the same arguments as in Eq. (40), it follows that the CMSM is a weighted GMSM with weight function $q(\mathbf{a}, \mathbf{x}) = 0$.

### C.3 Longitudinal marginal sensitivity model (LMSM):

For longitudinal settings with time-varying observed confounders $\mathbf{X} = \bar{\mathbf{X}}_T = (\mathbf{X}_1, \ldots, \mathbf{X}_T)$, unobserved confounders $\mathbf{U} = \bar{\mathbf{U}}_T = (\mathbf{U}_1, \ldots, \mathbf{U}_T)$, treatments $\mathbf{A} = \bar{\mathbf{A}}_T = (\mathbf{A}_1, \ldots, \mathbf{A}_T)$, we define the longitudinal marginal sensitivity model (LMSM) via

$$\frac{1}{\Gamma} \leq \prod_{t=1}^{T} \frac{\mathbb{P}(\mathbf{a}_t \mid \bar{\mathbf{x}}_T, \bar{\mathbf{u}}_t, \bar{\mathbf{a}}_{t-1})}{\mathbb{P}(\mathbf{a}_t \mid \bar{\mathbf{x}}_T, \bar{\mathbf{a}}_{t-1})} \leq \Gamma. \tag{45}$$

It holds that

$$\mathbb{P}(\bar{\mathbf{u}}_T \mid \bar{\mathbf{x}}_T, \bar{\mathbf{a}}_T) = \frac{\prod_{t=1}^{T} \mathbb{P}(\bar{\mathbf{u}}_t \mid \bar{\mathbf{x}}_T, \bar{\mathbf{a}}_t)}{\prod_{t=1}^{T-1} \mathbb{P}(\bar{\mathbf{u}}_t \mid \bar{\mathbf{x}}_T, \bar{\mathbf{a}}_t)} = \prod_{t=1}^{T} \frac{\mathbb{P}(\bar{\mathbf{u}}_t \mid \bar{\mathbf{x}}_T, \bar{\mathbf{a}}_t)}{\mathbb{P}(\bar{\mathbf{u}}_{t-1} \mid \bar{\mathbf{x}}_T, \bar{\mathbf{a}}_{t-1})} \tag{46}$$

$$\overset{(*)}{=} \left( \prod_{t=1}^{T} \frac{\mathbb{P}(\mathbf{a}_t \mid \bar{\mathbf{x}}_T, \bar{\mathbf{u}}_t, \bar{\mathbf{a}}_{t-1})}{\mathbb{P}(\mathbf{a}_t \mid \bar{\mathbf{x}}_T, \bar{\mathbf{a}}_{t-1})} \right) \left( \prod_{t=1}^{T} \frac{\mathbb{P}(\bar{\mathbf{u}}_t \mid \bar{\mathbf{x}}_T, \bar{\mathbf{a}}_{t-1})}{\mathbb{P}(\bar{\mathbf{u}}_{t-1} \mid \bar{\mathbf{x}}_T, \bar{\mathbf{a}}_{t-1})} \right) \tag{47}$$

$$= \left( \prod_{t=1}^{T} \frac{\mathbb{P}(\mathbf{a}_t \mid \bar{\mathbf{x}}_T, \bar{\mathbf{u}}_t, \bar{\mathbf{a}}_{t-1})}{\mathbb{P}(\mathbf{a}_t \mid \bar{\mathbf{x}}_T, \bar{\mathbf{a}}_{t-1})} \right) \left( \prod_{t=1}^{T} \mathbb{P}(\mathbf{u}_t \mid \bar{\mathbf{x}}_T, \bar{\mathbf{a}}_{t-1}, \bar{\mathbf{u}}_{t-1}) \right) \tag{48}$$

$$= \left( \prod_{t=1}^{T} \frac{\mathbb{P}(\mathbf{a}_t \mid \bar{\mathbf{x}}_T, \bar{\mathbf{u}}_t, \bar{\mathbf{a}}_{t-1})}{\mathbb{P}(\mathbf{a}_t \mid \bar{\mathbf{x}}_T, \bar{\mathbf{a}}_{t-1})} \right) \mathbb{P}(\bar{\mathbf{u}}_T \mid \bar{\mathbf{x}}_T, do(\bar{\mathbf{A}}_T = \bar{\mathbf{a}}_T)), \tag{49}$$

where $(*)$ follows by applying Bayes' theorem on $\mathbb{P}(\bar{\mathbf{u}}_t \mid \bar{\mathbf{x}}_T, \bar{\mathbf{a}}_t)$. Hence, the LMSM is a weighted GMSM with weight function $q(\bar{\mathbf{a}}_T, \bar{\mathbf{x}}_T) = 0$. Note that we can also define LMSMs with different weight functions via

$$\frac{1}{(1 - \Gamma)q(\bar{\mathbf{a}}_T, \bar{\mathbf{x}}_T) + \Gamma} \leq \prod_{t=1}^{T} \frac{\mathbb{P}(\mathbf{a}_t \mid \bar{\mathbf{x}}_T, \bar{\mathbf{u}}_t, \bar{\mathbf{a}}_{t-1})}{\mathbb{P}(\mathbf{a}_t \mid \bar{\mathbf{x}}_T, \bar{\mathbf{a}}_{t-1})} \leq \frac{1}{(1 - \Gamma^{-1})q(\bar{\mathbf{a}}_T, \bar{\mathbf{x}}_T) + \Gamma^{-1}}. \tag{50}$$

# D  Bounds for average causal effects and differences

Here, we show that we can use our sharp bounds to obtain sharp bounds for causal effect averages and differences. We state the results for the upper bound

$$Q^+(\mathbf{x}, \bar{\mathbf{a}}_{\ell+1}, \mathcal{S}) = \sup_{\mathcal{M} \in \mathcal{C}(\mathcal{S})} Q(\mathbf{x}, \bar{\mathbf{a}}_{\ell+1}, \mathcal{M}). \tag{51}$$

All definitions and bounds for the lower bound $Q^-(\mathbf{x}, \bar{\mathbf{a}}_{\ell+1}, \mathcal{S})$ can be obtained by swapping the signs.

We are interested in the sharp upper bound for the *average causal effect*

$$Q^+_{\text{avg}}(\bar{\mathbf{a}}_{\ell+1}, \mathcal{S}) = \sup_{\mathcal{M} \in \mathcal{C}(\mathcal{S})} \int_{\mathcal{X}} Q(\mathbf{x}, \bar{\mathbf{a}}_{\ell+1}, \mathcal{M}) \, \mathrm{d}\mathbf{x} \tag{52}$$

and the sharp upper bound for the *causal effect difference*

$$Q^+_{\text{diff}}\left(\mathbf{x}, \bar{\mathbf{a}}^{(1)}_{\ell+1}, \bar{\mathbf{a}}^{(2)}_{\ell+1}, \mathcal{S}\right) = \sup_{\mathcal{M} \in \mathcal{C}(\mathcal{S})} \left(Q\left(\mathbf{x}, \bar{\mathbf{a}}^{(1)}_{\ell+1}, \mathcal{M}\right) - Q\left(\mathbf{x}, \bar{\mathbf{a}}^{(2)}_{\ell+1}, \mathcal{M}\right)\right). \tag{53}$$

**Lemma 3.** *We can compute* $Q^+_{\text{avg}}(\bar{\mathbf{a}}_{\ell+1}, \mathcal{S})$ *and* $Q^+_{\text{diff}}\left(\mathbf{x}, \bar{\mathbf{a}}^{(1)}_{\ell+1}, \bar{\mathbf{a}}^{(2)}_{\ell+1}, \mathcal{S}\right)$ *from our sharp bounds* $Q^+(\mathbf{x}, \bar{\mathbf{a}}_{\ell+1}, \mathcal{S})$ *and* $Q^-(\mathbf{x}, \bar{\mathbf{a}}_{\ell+1}, \mathcal{S})$ *via*

$$Q^+_{\text{avg}}(\bar{\mathbf{a}}_{\ell+1}, \mathcal{S}) = \int_{\mathcal{X}} Q^+(\mathbf{x}, \bar{\mathbf{a}}_{\ell+1}, \mathcal{S}) \, \mathrm{d}\mathbf{x} \tag{54}$$

*and*

$$Q^+_{\text{diff}}\left(\mathbf{x}, \bar{\mathbf{a}}^{(1)}_{\ell+1}, \bar{\mathbf{a}}^{(2)}_{\ell+1}, \mathcal{S}\right) = Q^+(\mathbf{x}, \bar{\mathbf{a}}^{(1)}_{\ell+1}, \mathcal{S}) - Q^-(\mathbf{x}, \bar{\mathbf{a}}^{(2)}_{\ell+1}, \mathcal{S}). \tag{55}$$

*Proof.* The result for $Q^+_{\text{avg}}(\bar{\mathbf{a}}_{\ell+1}, \mathcal{S})$ follows directly from interchanging the supremum and integral. For $Q^+_{\text{diff}}\left(\mathbf{x}, \bar{\mathbf{a}}^{(1)}_{\ell+1}, \bar{\mathbf{a}}^{(2)}_{\ell+1}, \mathcal{S}\right)$, we note that

$$Q^+_{\text{diff}}\left(\mathbf{x}, \bar{\mathbf{a}}^{(1)}_{\ell+1}, \bar{\mathbf{a}}^{(2)}_{\ell+1}, \mathcal{S}\right) \leq \sup_{\mathcal{M}_1 \in \mathcal{C}(\mathcal{S})} Q(\mathbf{x}, \bar{\mathbf{a}}^{(1)}_{\ell+1}, \mathcal{M}_1) - \inf_{\mathcal{M}_2 \in \mathcal{C}(\mathcal{S})} Q(\mathbf{x}, \bar{\mathbf{a}}^{(2)}_{\ell+1}, \mathcal{M}_2) \tag{56}$$

$$= Q^+(\mathbf{x}, \bar{\mathbf{a}}^{(1)}_{\ell+1}, \mathcal{S}) - Q^-(\mathbf{x}, \bar{\mathbf{a}}^{(2)}_{\ell+1}, \mathcal{S}). \tag{57}$$

To show the equality in Eq. (56), we show that, for each pair of SCMs $\mathcal{M}_1, \mathcal{M}_2 \in \mathcal{C}(\mathcal{S})$, we can find an SCM $\mathcal{M} \in \mathcal{C}(\mathcal{S})$ such that

$$Q(\mathbf{x}, \bar{\mathbf{a}}^{(1)}_{\ell+1}, \mathcal{M}_1) - Q(\mathbf{x}, \bar{\mathbf{a}}^{(2)}_{\ell+1}, \mathcal{M}_2) = Q(\mathbf{x}, \bar{\mathbf{a}}^{(1)}_{\ell+1}, \mathcal{M}) - Q(\mathbf{x}, \bar{\mathbf{a}}^{(2)}_{\ell+1}, \mathcal{M}). \tag{58}$$

We can assume w.l.o.g. that all $\mathcal{M} \in \mathcal{C}(\mathcal{S})$ induce the same distributions $\mathbb{P}^{\mathbf{U}_W | \mathbf{x}}$ on the confounding space. We denote the functional assignments of $\mathcal{M}_1$ and $\mathcal{M}_2$ as $f^W_{\mathcal{M}_1}(\mathbf{x}, \mathbf{m}_W, \mathbf{a}, \mathbf{u}_W)$ and $f^W_{\mathcal{M}_2}(\mathbf{x}, \mathbf{m}_W, \mathbf{a}, \mathbf{u}_W)$. We can now define a functional assignment $f^W_{\mathcal{M}}(\mathbf{x}, \mathbf{m}_W, \mathbf{a}, \mathbf{u}_W)$ for $\mathcal{M}$ so that

$$f^W_{\mathcal{M}}(\cdot, \cdot, \mathbf{a}^{(1)}, \cdot) = f^W_{\mathcal{M}_1}(\cdot, \cdot, \mathbf{a}^{(1)}, \cdot) \quad \text{and} \quad f^W_{\mathcal{M}}(\cdot, \cdot, \mathbf{a}^{(2)}, \cdot) = f^W_{\mathcal{M}_2}(\cdot, \cdot, \mathbf{a}^{(2)}, \cdot), \tag{59}$$

which implies Eq. (58).

$\square$

# E   Compatibility of SCMs with sensitivity models

Here, we provide details regarding our class $\mathcal{C}(\mathcal{S})$ of SCMs that are compatible with a sensitivity model $\mathcal{S}$ from Def. 2.

**Definition 6** (Compatibility)**.** Let $\mathbb{P}^{\mathbf{V}}$ denote the distribution of the observed variables $\mathbf{V} = (\mathbf{X}, \mathbf{A}, \bar{\mathbf{M}}_\ell, Y)$ and let $\mathcal{S}$ be a sensitivity model with unobserved confounders $\mathbf{U}$ and a family of joint distributions $\mathcal{P}$. Let $\mathcal{M}$ be an SCM with observed variables $\mathbf{V}_{\mathcal{M}} = \mathbf{V}$ and unobserved variables $\mathbf{U}_{\mathcal{M}} \supseteq \mathbf{U}$ that contain the unobserved confounders $\mathbf{U}$ from the sensitivity model but also potential exogenous noise. Then, $\mathcal{M}$ is compatible with $\mathcal{S}$, if the following three conditions are satisfied:

1. Graph compatibility: the induced causal graph $\mathcal{G}_{\mathcal{M}}$ by $\mathcal{M}$ on $\mathbf{V} \cup \mathbf{U}$ must coincide with our assumed causal structure (see Sec. 3.1).

2. Latent unconfoundedness: $\mathbb{P}(w \mid \mathbf{x}, \mathbf{m}_W, do(\mathbf{A} = \mathbf{a})) = \int \mathbb{P}(w \mid \mathbf{x}, \mathbf{u}_W, \mathbf{m}_W, \mathbf{a}) \mathbb{P}(\mathbf{u}_W \mid \mathbf{x}, \mathbf{m}_W) \, \mathrm{d}\mathbf{u}_W$ for all $\mathbf{x} \in \mathcal{X}$, $\mathbf{m}_W \in supp(\mathbf{M}_W)$, $\mathbf{a} \in \mathcal{A}$, and $W \in \{M_1, \ldots, M_\ell, Y\}$.

3. Sensitivity constraints: $\mathbb{P}_{\mathcal{M}}^{\mathbf{V} \cup \mathbf{U}} \in \mathcal{P}$, where $\mathbb{P}_{\mathcal{M}}^{\mathbf{V} \cup \mathbf{U}}$ denotes the distribution on $\mathbf{V} \cup \mathbf{U}$ induced by $\mathcal{M}$.

In Definition 6, the latent unconfoundedness condition ensures that all interventional densities $\mathbb{P}(w \mid \mathbf{x}, \mathbf{m}_W, do(\mathbf{A} = \mathbf{a}))$ are point-identified under access to the unobserved confounders. In particular, this ensures that the SCM $\mathcal{M}$ can not contain additional unobserved confounders (e.g., between mediators and outcome). Similar assumptions are standard in the causal sensitivity literature [18].

# F  Importance sampling estimators for finite sample bounds

In this section, we derive estimators for the outcome bound $\mathcal{D}\left(\mathbb{P}_+^Y(\cdot \mid \mathbf{x}, \bar{\mathbf{m}}_\ell, \mathbf{a})\right)$ for continuous $Y \in \mathbb{R}$. We assume that we have already obtained an estimator $\hat{\mathbb{P}}^Y(\cdot \mid \mathbf{x}, \bar{\mathbf{m}}_\ell, \mathbf{a})$ of the observational distribution $\mathbb{P}^Y(\cdot \mid \mathbf{x}, \bar{\mathbf{m}}_\ell, \mathbf{a})$, and that we are able to sample $(y_i)_{i=1}^k \sim \hat{\mathbb{P}}^Y(\cdot \mid \mathbf{x}, \bar{\mathbf{m}}_\ell, \mathbf{a})$ (see Appendix G for implementation details). Note that the outcome bound $\mathcal{D}\left(\mathbb{P}_+^Y(\cdot \mid \mathbf{x}, \bar{\mathbf{m}}_\ell, \mathbf{a})\right)$ depends on the shifted distribution $\mathbb{P}_+^Y(\cdot \mid \mathbf{x}, \bar{\mathbf{m}}_\ell, \mathbf{a})$ and not on the observational distribution $\mathbb{P}^Y(\cdot \mid \mathbf{x}, \bar{\mathbf{m}}_\ell, \mathbf{a})$. Hence, we use an importance sampling approach to derive our estimators, which we outline in the following for the expectation functional and distributional effects. We denote the CDFs corresponding to $\mathbb{P}_+^Y(\cdot \mid \mathbf{x}, \bar{\mathbf{m}}_\ell, \mathbf{a})$ and $\mathbb{P}^Y(\cdot \mid \mathbf{x}, \bar{\mathbf{m}}_\ell, \mathbf{a})$ by $F_{\mathbb{P}_+^Y \mid \mathbf{x}, \bar{\mathbf{m}}_\ell, \mathbf{a}}$ and $F_{\mathbb{P}^Y \mid \mathbf{x}, \bar{\mathbf{m}}_\ell, \mathbf{a}}$, respectively.

**Expectation functional:** For the expectation functional, we can rewrite the outcome bound as

$$\mathcal{D}\left(\mathbb{P}_+^Y(\cdot \mid \mathbf{x}, \bar{\mathbf{m}}_\ell, \mathbf{a})\right) = \mathbb{E}_{Y \sim \mathbb{P}_+^Y \mid \mathbf{x}, \bar{\mathbf{m}}_\ell, \mathbf{a}}[Y] \tag{60}$$

$$= \mathbb{E}_{Y \sim \mathbb{P}^Y \mid \mathbf{x}, \bar{\mathbf{m}}_\ell, \mathbf{a}}\left[Y \frac{\mathbb{P}_+^Y(Y \mid \mathbf{x}, \bar{\mathbf{m}}_\ell, \mathbf{a})}{\mathbb{P}^Y(Y \mid \mathbf{x}, \bar{\mathbf{m}}_\ell, \mathbf{a})}\right] \tag{61}$$

$$= \mathbb{E}_{Y \sim \mathbb{P}^Y \mid \mathbf{x}, \bar{\mathbf{m}}_\ell, \mathbf{a}}\left[\frac{Y}{s_Y^+} \mathbb{1}\left(Y \leq F_{\mathbb{P}^Y \mid \mathbf{x}, \bar{\mathbf{m}}_\ell, \mathbf{a}}^{-1}(c_Y^+)\right) + \frac{Y}{s_Y^-} \mathbb{1}\left(Y > F_{\mathbb{P}^Y \mid \mathbf{x}, \bar{\mathbf{m}}_\ell, \mathbf{a}}^{-1}(c_Y^+)\right)\right] \tag{62}$$

to obtain the consistent estimator

$$\mathcal{D}\left(\widehat{\mathbb{P}_+^Y(\cdot \mid \mathbf{x}}, \bar{\mathbf{m}}_\ell, \mathbf{a})\right) = \frac{1}{k} \sum_{i=1}^{\lfloor kc_Y^+ \rfloor} \frac{y_i}{\hat{s}_Y^+} + \frac{1}{k} \sum_{i=\lfloor kc_Y^+ \rfloor + 1}^{k} \frac{y_i}{\hat{s}_Y^-}, \tag{63}$$

where $(y_i)_{i=1}^k \sim \hat{\mathbb{P}}^Y(\cdot \mid \mathbf{x}, \bar{\mathbf{m}}_\ell, \mathbf{a})$ are sampled from the estimated observational distribution. This corresponds to Eq. (8) in the main paper.

**Computational complexity:** Given trained models and a sample $(y_i)_{i=1}^k \sim \hat{\mathbb{P}}^Y(\cdot \mid \mathbf{x}, \bar{\mathbf{m}}_\ell, \mathbf{a})$, our estimator from Eq. (63) has a complexity of $\mathcal{O}(k)$ as it only involves summing and quantile computation. To set this in relation to existing work, Algorithm 1 of Jesson et al. [32] has a complexity of $\mathcal{O}(kn)$, where $n$ is a number of grid search points. This is demonstrated in Table 1, where we choose $k = n = 5000$.

**Distributional effects:** We now derive estimators for distributional effects, i.e., for quantile functionals $\mathcal{D}$ of the form

$$\mathcal{D}\left(\mathbb{P}_+^Y(\cdot \mid \mathbf{x}, \bar{\mathbf{m}}_\ell, \mathbf{a})\right) = F_{\mathbb{P}_+^Y \mid \mathbf{x}, \bar{\mathbf{m}}_\ell, \mathbf{a}}^{-1}(\alpha) \tag{64}$$

with $\alpha \in (0,1)$. We again use an importance sampling approach and rewrite

$$F_{\mathbb{P}_+^Y \mid \mathbf{x}, \bar{\mathbf{m}}_\ell, \mathbf{a}}(y) = \mathbb{E}_{Y \sim \mathbb{P}_+^Y \mid \mathbf{x}, \bar{\mathbf{m}}_\ell, \mathbf{a}}[\mathbb{1}(Y \leq y)] \tag{65}$$

$$= \mathbb{E}_{Y \sim \mathbb{P}^Y \mid \mathbf{x}, \bar{\mathbf{m}}_\ell, \mathbf{a}}\left[\mathbb{1}(Y \leq y) \frac{\mathbb{P}_+^Y(Y \mid \mathbf{x}, \bar{\mathbf{m}}_\ell, \mathbf{a})}{\mathbb{P}^Y(Y \mid \mathbf{x}, \bar{\mathbf{m}}_\ell, \mathbf{a})}\right] \tag{66}$$

$$= \mathbb{E}_{Y \sim \mathbb{P}^Y \mid \mathbf{x}, \bar{\mathbf{m}}_\ell, \mathbf{a}}\left[\frac{\mathbb{1}\left(Y \leq \min\{y, F_{\mathbb{P}^Y \mid \mathbf{x}, \bar{\mathbf{m}}_\ell, \mathbf{a}}^{-1}(c_Y^+)\}\right)}{s_Y^+} + \frac{\mathbb{1}\left(F_{\mathbb{P}^Y \mid \mathbf{x}, \bar{\mathbf{m}}_\ell, \mathbf{a}}^{-1}(c_Y^+) < Y \leq y\right)}{s_Y^-}\right]. \tag{67}$$

Hence, we can sample $(y_i)_{i=1}^k \sim \hat{\mathbb{P}}^Y(\cdot \mid \mathbf{x}, \bar{\mathbf{m}}_\ell, \mathbf{a})$ and obtain the consistent estimator

$$\mathcal{D}\left(\widehat{\mathbb{P}_+^Y(\cdot \mid \mathbf{x}}, \bar{\mathbf{m}}_\ell, \mathbf{a})\right) = \min_{\hat{F}_{\mathbb{P}_+^Y \mid \mathbf{x}, \bar{\mathbf{m}}_\ell, \mathbf{a}}(y_i) \geq \alpha} y_i, \tag{68}$$

where

$$\hat{F}_{\mathbb{P}_+^Y \mid \mathbf{x}, \bar{\mathbf{m}}_\ell, \mathbf{a}}(y) = \frac{1}{k} \sum_{i=1}^{\lfloor kc_Y^+ \rfloor} \frac{\mathbb{1}(y_i \leq y)}{\hat{s}_Y^+} + \frac{1}{k} \sum_{i=\lfloor kc_Y^+ \rfloor + 1}^{k} \frac{\mathbb{1}(y_i \leq y)}{\hat{s}_Y^-}. \tag{69}$$

# G   Implementation and hyperparameter tuning details

All our experimental settings feature a continuous outcome $Y \in \mathbb{R}$ and (optionally) discrete mediators $M_i \in \mathbb{N}$. Hence, we need to estimate the conditional outcome density $\mathbb{P}^Y(\cdot \mid \mathbf{x}, \mathbf{m}, \mathbf{a})$ and conditional probability mass functions $\mathbb{P}^{M_i}(\cdot \mid \mathbf{x}, \bar{\mathbf{m}}_{i-1}, \mathbf{a})$ in order to estimate our bounds with Eq. (63), Eq. (68), and Algorithm 1 from the main paper.

**Conditional outcome density:** We use conditional normalizing flows (CNFs) [73] for estimating the conditional density $\mathbb{P}^Y(\cdot \mid \mathbf{x}, \bar{\mathbf{m}}_\ell, \mathbf{a})$. Normalizing flows (NFs) model a distribution $\mathbb{P}^Y$ of a target variable $Y$ by transforming a simple base distribution $\mathbb{P}^U$ (e.g., standard normal) of a latent variable $U$ through an invertible transformation $Y = f_\theta(U)$, where $\theta$ denotes learnable parameters [58]. In order to estimate the *conditional* density $\mathbb{P}^Y(\cdot \mid \mathbf{x}, \bar{\mathbf{m}}_\ell, \mathbf{a})$, we leverage CNFs, that is, we define the parameters $\theta$ as an output of a *hyper network* $\theta = g_\eta(\mathbf{x}, \bar{\mathbf{m}}_\ell, \mathbf{a})$ with learnable parameters $\eta$. Given a sample $\{\mathbf{x}_i, \bar{\mathbf{m}}_{\ell,i}, \mathbf{a}_i, y_i\}_{i=1}^n$, we learn $\eta$ by maximizing the log-likelihood

$$\ell(\eta) = \sum_{i=1}^n \log \left( f_{g_\eta(\mathbf{x}_i, \bar{\mathbf{m}}_{\ell,i}, \mathbf{a}_i) \#} \mathbb{P}^U(y_i) \right) \tag{70}$$

$$\overset{(*)}{=} \sum_{i=1}^n \log \left( \mathbb{P}^U \left( f_{g_\eta(\mathbf{x}_i, \bar{\mathbf{m}}_{\ell,i}, \mathbf{a}_i)}^{-1}(y_i) \right) \right) + \log \left( \left| \frac{\mathrm{d}}{\mathrm{d}y} f_{g_\eta(\mathbf{x}_i, \bar{\mathbf{m}}_{\ell,i}, \mathbf{a}_i)}^{-1}(y_i) \right| \right), \tag{71}$$

where $f_{g_\eta(\mathbf{x}_i, \bar{\mathbf{m}}_{\ell,i}, \mathbf{a}_i) \#} \mathbb{P}^U(y_i)$ denotes the (push-forward) density induced by $f_{g_\eta(\mathbf{x}_i, \bar{\mathbf{m}}_{\ell,i}, \mathbf{a}_i)}$ on $\mathbb{R}$ and $(*)$ follows from the change-of-variables theorem for invertible transformations.

In our implementation, we use neural spline flows. That is, we model the invertible transformation $f_\theta$ via a spline flow as described in [21]. We use a feed-forward neural network for the hypernetwork $g_\eta(\mathbf{x}, \bar{\mathbf{m}}_\ell, \mathbf{a})$ with 2 hidden layers, ReLU activation functions, and linear output. We set the latent distribution $\mathbb{P}^U$ to a standard normal distribution $\mathcal{N}(0, 1)$. For training, we use the Adam optimizer [41].

**Conditional probability mass functions:** The estimation of the conditional probability mass function $\mathbb{P}^{M_i}(\cdot \mid \mathbf{x}, \bar{\mathbf{m}}_{i-1}, \mathbf{a})$ is a standard (multi-class) classification problem. We use feed-forward neural networks with 3 hidden layers, ReLU activation functions, and softmax output. For training, we minimize the standard cross-entropy loss by using the Adam optimizer [41]. We use the same approach to estimate the propensity scores $\mathbb{P}^{\mathbf{A}}(\cdot \mid \mathbf{x})$ for discrete treatments $\mathbf{A}$.

**Hyperparameter tuning:**

We perform hyperparameter tuning for our experiments on synthetic data using grid search on a validation set. The tunable parameters and search ranges are shown in Table 2. For reproducibility purposes, we report the selected hyperparameters as *.yaml* files.[3]

Table 2: Hyperparameter tuning details.

| MODEL | TUNABLE PARAMETERS | SEARCH RANGE |
|---|---|---|
| CNFs | Epochs | 50 |
| | Batch size | 32, 64, 128 |
| | Learning rate | 0.0005, 0.001, 0.005 |
| | Hidden layer size (hyper network) | 5, 10, 20, 30 |
| | Number of spline bins | 2, 4, 8 |
| Feed forward neural networks | Epochs | 30 |
| | Batch size | 32, 64, 128 |
| | Learning rate | 0.0005, 0.001, 0.005 |
| | Hidden layer size | 5, 10, 20, 30 |
| | Dropout probability | 0, 0.1 |

---

[3] Code is available in the supplementary materials and at https://github.com/DennisFrauen/SharpCausalSensitivity.

# H  Experiments using synthetic data

Here we provide details regarding our experiments using synthetic data. This includes data generation, obtaining oracle sensitivity parameters, and details regarding experimental evaluation.

**Overall data-generating process:** We first describe the overall data-generating process which we use as a basis to generate data for all settings (i)-(iii) and binary/continuous treatments. We construct an SCM following the causal graph in Fig. 1 (right) from the main paper. We have an observed confounder $X \in \mathbb{R}$, a (binary or continuous) treatment $A$, two binary mediators $M_1$ and $M_2$, and a continuous outcome $Y \in \mathbb{R}$. Furthermore, we consider three unobserved confounders: (i) $U_{M_1}$ confounding the $A$-$M_1$ relationship, (ii) $U_{M_2}$ confounding the $A$–$M_2$ relationship, and (iii) $U_Y$ confounding the $A$–$Y$ relationship. Our data-generating process is inspired by synthetic experiments from previous works on causal sensitivity analysis [31, 35]. We start the data-generating process by sampling

$$X \sim \text{Uniform}[-1, 1], \quad \text{and} \quad U_{M_1}, U_{M_2}, U_Y \overset{(i.i.d)}{\sim} \text{Bernoulli}(p = 0.5) \tag{72}$$

Depending on the setting, we either generate binary treatments $A \in \{0, 1\}$ via

$$A \sim \text{Bernoulli}(\text{sigmoid}(3\text{x} + \gamma_{M_1} u_{M_1} + \gamma_{M_2} u_{M_2} + \gamma_Y u_Y)) \tag{73}$$

or continuous treatments $A \in (0, 1)$ via

$$A \sim \text{Beta}(\alpha, \beta) \text{ with } \alpha = \beta = 2 + x + \gamma_{M_1}(u_{M_1} - 0.5) + \gamma_{M_2}(u_{M_2} - 0.5) + \gamma_Y(u_Y - 0.5), \tag{74}$$

where $\gamma_{M_1}$, $\gamma_{M_2}$, and $\gamma_Y$ are parameters controlling the strength of unobserved confounding. We then generate the mediators and outcome via functional assignments

$$M_1 = f_{M_1}(X, A, U_{M_1}, \epsilon_{M_1}), \quad M_2 = f_{M_2}(X, A, M_1, U_{M_2}, \epsilon_{M_2}) \tag{75}$$

and

$$Y = f_Y(X, A, M_1, M_2 U_Y, \epsilon_Y), \tag{76}$$

where $\epsilon_{M_1}, \epsilon_{M_2}, \epsilon_Y \sim \mathcal{N}(0, 1)$ are standard normal distributed noise variables. The functional assignments are defined as

$$f_{M_1}(x, a, u_{M_1}, \epsilon_{M_1}) = \mathbb{1}\{a \sin(x) + (1 - a)\sin(4x) + \rho_{M_1}((u_{M_1} - 0.5) + \epsilon_{M_1}) > 0\} \tag{77}$$

for $M_1$,

$$f_{M_2}(x, a, m_1 u_{M_2}, \epsilon_{M_2}) = \mathbb{1}\{a\, m_1 \sin(x) + (1 - a)\, m_1 \sin(4x) \tag{78}$$
$$- a(1 - m_1)\sin(x) - (1 - a)(1 - m_1)\sin(4x) \tag{79}$$
$$+ \rho_{M_2}((u_{M_2} - 0.5) + \epsilon_{M_2}) > 0\} \tag{80}$$

for $M_2$, and

$$f_Y(x, a, m_1, m_2, u_Y, \epsilon_Y) = a\, m_1 m_2 \sin(x) + (1 - a)\, m_1 m_2 \sin(4x) \tag{81}$$
$$+ a\, m_1(1 - m_2)\sin(8x) + (1 - a)\, m_1(1 - m_2)\sin(x) \tag{82}$$
$$- a(1 - m_1) m_2 \sin(x) - (1 - a)(1 - m_1) m_2 \sin(4x) \tag{83}$$
$$- a(1 - m_1)(1 - m_2)\sin(8x) \tag{84}$$
$$- (1 - a)(1 - m_1)(1 - m_2)\sin(x) \tag{85}$$
$$+ \rho_Y((u_Y - 0.5) + \epsilon_Y) \tag{86}$$

for $Y$, where $\rho_{M_1}$, $\rho_{M_2}$, and $\rho_Y$ are parameters that control the noise level.

**Settings (i)-(iii):** We define the settings (i)-(iii) in Sec. 5 via specific values of the confounding parameters $\gamma_{M_1}$, $\gamma_{M_2}$, and $\gamma_Y$, and the noise parameters $\rho_{M_1}$, $\rho_{M_2}$, and $\rho_Y$ (see Table 3). Note that the settings are defined to mimic the causal graphs in Fig. 1 from the main paper. For example, the only unobserved confounder in setting (i) is $U_Y$, which means that we can ignore the mediators and use our data to evaluate our bounds for settings without mediators.

Table 3: Definition of settings (i)-(iii).

| | $\gamma_{M_1}$ | $\gamma_{M_2}$ | $\gamma_Y$ | $\rho_{M_1}$ | $\rho_{M_2}$ | $\rho_Y$ |
|---|---|---|---|---|---|---|
| Setting (i), binary $A$ | 0 | 0 | 1.5 | 0.2 | 0.2 | 2 |
| Setting (i), continuous $A$ | 0 | 0 | 1.5 | 0.2 | 0.2 | 1 |
| Setting (ii) | 1.5 | 0 | 1.5 | 1 | 0.2 | 1 |
| Setting (iii) | 1.5 | 1.5 | 1.5 | 0.2 | 0.2 | 1 |

**Obtaining $\Gamma_W^*$:** We provide details regarding our approach to obtain oracle sensitivity parameters $\Gamma_W^*$ for all $W \in \{M_1, M_2, Y\}$. By sampling from our previously defined SCM, we can obtain Monte Carlo estimates of the GMSM density ratio

$$r(u_W, x, a) = \frac{\mathbb{P}(u_W \mid x, a)}{\mathbb{P}(u_W \mid x, a)} \overset{(*)}{=} \frac{\mathbb{P}(a \mid x, u_W)}{\mathbb{P}(a \mid x)} \tag{87}$$

for all $u_W \in \{0, 1\}$, $a$, and $x$, where $(*)$ follows from Bayes' theorem. We then define

$$r_W^+(x, a) = \max_{u_W \in \{0,1\}} r(u_W, x, a) \quad \text{and} \quad r_W^-(x, a) = \min_{u_W \in \{0,1\}} r(u_W, x, a). \tag{88}$$

For binary treatment settings, we define parameters $\Gamma_W^+ = \Gamma_W^+(x, a)$ and $\Gamma_W^- = \Gamma_W^-(x, a)$ that attain the density ratio bounds in the MSM from Eq. (39), i.e.

$$r_W^+(x, a) = \frac{1}{(1 - \Gamma_W^{+^{-1}})\mathbb{P}(a \mid x) + \Gamma_W^{+^{-1}}} \quad \text{and} \quad r_W^-(x, a) = \frac{1}{(1 - \Gamma_W^-)\mathbb{P}(a \mid x) + \Gamma_W^-}. \tag{89}$$

For continuous treatment settings, we define $\Gamma_W^+$ and $\Gamma_W^-$ as the sensitivity parameters that attain the density ratio bounds in the CMSM from Eq. (44), i.e.

$$r_W^+(x, a) = \Gamma_W^+ \quad \text{and} \quad r_W^-(x, a) = \frac{1}{\Gamma_W^-}. \tag{90}$$

Finally, we define $\Gamma_W^*$ as the parameter corresponding to the maximum possible violation of unconfoundedness, i.e.,

$$\Gamma_W^* = \max\{\Gamma_W^+, \Gamma_W^-\} \tag{91}$$

By definition of $\Gamma_W^*$, our bounds should contain the oracle causal effect whenever we choose sensitivity parameters $\Gamma_W \geq \Gamma_W^*$ for all $W \in \{M_1, M_2, Y\}$.

**Weighted GMSM experiment (Table 1):** For our experiment in Table 1, we modify the treatment assignment from Eq. (74) in setting (i) to

$$A \sim \text{Beta}(\alpha, \beta) \tag{92}$$

with

$$\alpha = \beta = 2 + x + \mathbb{1}(x < 0)\left(\gamma_{M_1}(u_{M_1} - 0.5) + \gamma_{M_2}(u_{M_2} - 0.5) + \gamma_Y(u_Y - 0.5)\right). \tag{93}$$

Hence, unobserved confounding only affects individuals with $x < 0$. We then compare our bounds under the CMSM with our bounds under a weighted CMSM (Def. 4) with weight function $q_Y(x) = \mathbb{1}(x > 0)$.

We also provide results for the bounds from Jesson et al. [32] under the CMSM. We implemented the grid search algorithm from Jesson et al. [32] and used $5,000$ samples for the search space. For a fair comparison, we also used $5,000$ samples for our importance sampling estimators. Note that the method from Jesson et al. [32] requires estimation of both the conditional outcome density $\mathbb{P}^Y(\cdot \mid \mathbf{x}, \mathbf{a})$ and the conditional expectation $\mathbb{E}[Y \mid \mathbf{x}, \mathbf{a}]$. For $\mathbb{P}^Y(\cdot \mid \mathbf{x}, \mathbf{a})$, we use the same (normalizing flow-based) estimator as for our bounds. For $\mathbb{E}[Y \mid \mathbf{x}, \mathbf{a}]$, we train a separate feed-forward neural network with linear output activation for continuous outcomes. Implementation and hyperparameter tuning are done the same way as described in Appendix G for the feed-forward neural networks.

# I   Experiment using real-world data

**Data:** We consider a setting from the COVID-19 pandemic where mobility in Switzerland (captured through telephone movement) was monitored to obtain a leading predictor of case growth. In total, $\sim 1,5$ billion trips were monitored from 10 February through 26 April 2020. All data are recorded across 26 different states (cantons). For our analysis, we use an aggregated, de-identified, and pre-processed version of the data provided by Persson, Parie, and Feuerriegel [57]. The preprocessed data is publically available at https://github.com/jopersson/covid19-mobility/blob/main/Data. The code for our analysis is available at https://github.com/DennisFrauen/SharpCausalSensitivity.

We consider a binary treatment $A$ in the form of a stay-at-home order, which bans gatherings with more than 5 people. We encode mobility as a single binary mediator $M$, which is 1 if the total number of trips on a specific day is larger than the median number of trips during the entire time horizon, and 0 otherwise. Our outcome is the 10-day-ahead case growth. We include the following observed variables as confounders $\mathbf{X}$: the canton code (swiss member state at a subnational level), the canton population, and whether the weekday is a Monday or not. After removing the first 10 recorded days for each canton (due to spillover effects from other countries) and rows with missing values, we obtain a dataset with $n = 3276$ observations.

**Analysis:** We perform a causal sensitivity analysis for the natural directed effect (NDE) of the stay-at-home order $A$ on the case growth $Y$. That is, we are interested in the part of the causal effect of $A$ on $Y$ that is not explained by the path via $M$ (i.e., through the change in mobility). The NDE in an SCM $\mathcal{M}$ is defined as

$$NDE(\mathcal{M}) = \int Q(\mathbf{x}, (a_1 = 0, a_2 = 1), \mathcal{M}) - Q(\mathbf{x}, (a_1 = 0, a_2 = 0), \mathcal{M}) \, \mathrm{d}\mathbf{x}. \tag{94}$$

Fig. 5 (main paper) shows causal sensitivity analysis for violations of the unconfoundedness between treatment $A$ and mediator $M$. Hence, we consider a GMSM for binary treatments with sensitivity parameters $\Gamma_M$ and $\Gamma_Y = 0$. For each $\Gamma_M$, we estimate our bounds for the expectation functional and the treatment combinations $\bar{\mathbf{a}} = (0, 1)$ and $\bar{\mathbf{a}} = (0, 0)$. We then obtain bounds for the NDE as described in Appendix D.

# J   Additional experimental results

Here, we provide additional experimental results on synthetic data that extend the results from Sec. 5 in the main paper. We provide (i) results for additional treatment combinations and (ii) results for distributional effects. We follow the same experimental setup described in Sec. 5 (main paper) and Appendix. H.

## J.1   Additional treatment combinations

Results for additional treatment combinations are shown in Fig. 6 (binary treatment settings) and Fig. 7 (continuous treatment settings). The results are similar to those in Sec. 5 in the main paper and empirically confirm the validity of our bounds. Hence, our results remain valid independently of the choice of treatment combination.

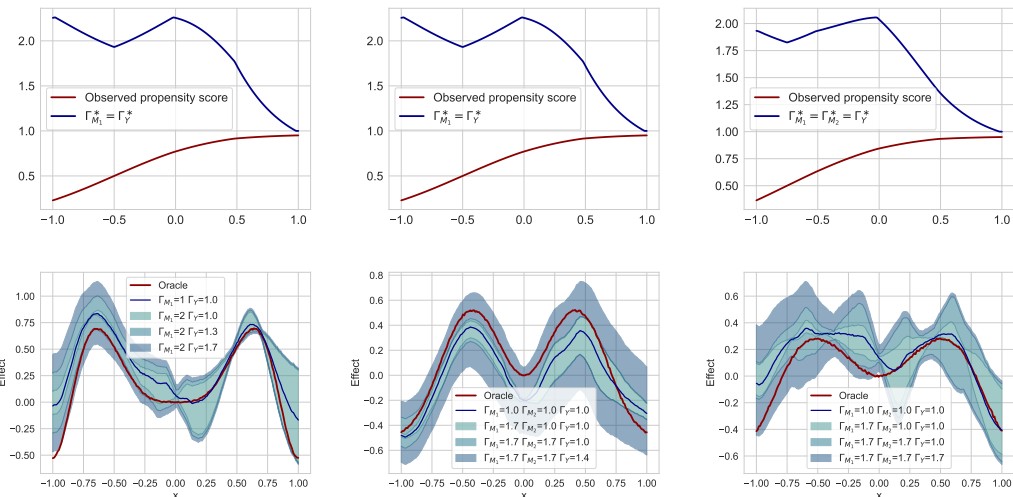

Figure 6: Results for additional treatments in the binary treatment setting. From left to right is shown: setting (ii) with $\bar{\mathbf{a}} = (0, 1)$, setting (iii) with $\bar{\mathbf{a}} = (0, 1, 0)$, and setting (iii) with $\bar{\mathbf{a}} = (0, 0, 1)$. The top row shows the oracle sensitivity parameter $\Gamma_W^*$ (depending on $x$), and the bottom row shows the bounds.

## J.2   Distributional effects

We also provide results for distributional effects, that is, we choose the $\alpha$-quantile functional $\mathcal{D}\left(\mathbb{P}_+^Y(\cdot \mid \mathbf{x}, \bar{\mathbf{m}}_\ell, \mathbf{a})\right) = F_{\mathbb{P}_+^Y \mid \mathbf{x}, \bar{\mathbf{m}}_\ell, \mathbf{a}}^{-1}(\alpha)$. Here, we consider three quantiles with $\alpha = 0.7$, $\alpha = 0.5$ (median), and $\alpha = 0.3$. We use our importance sampling estimator derived in Appendix. F (Eq. (68)) to estimate our bounds. The results are shown in Fig. 8 (binary treatment) and Fig. 9 (continuous treatment) for settings (i)-(iii) from Fig. 1 in the main paper. Again, our bounds cover the underlying oracle effect in regions where the chosen sensitivity parameters $\Gamma_W$ are larger than the oracle sensitivity parameters $\Gamma_W^*$. This also confirms empirically the validity of our bounds for distributional effects.

## J.3   Semi-synthetic data

We provide additional results for the semi-synthetic IHDP data with unobserved confounding from Jesson et al. [31]. Here, we demonstrate the effectiveness of our bounds for distributional effects for decision-making. We follow Jesson et al. [31] and evaluate our bounds by measuring the error rate of associated deferral policies, which defer a prespecified fraction of test samples with the largest uncertainty to an expert. We mimic a high-stakes decision-making problem where a wrong decision to prescribe treatment is much worse than a wrong decision not to prescribe treatment. To do so, we measure performance with a weighted error rate, that penalizes wrong

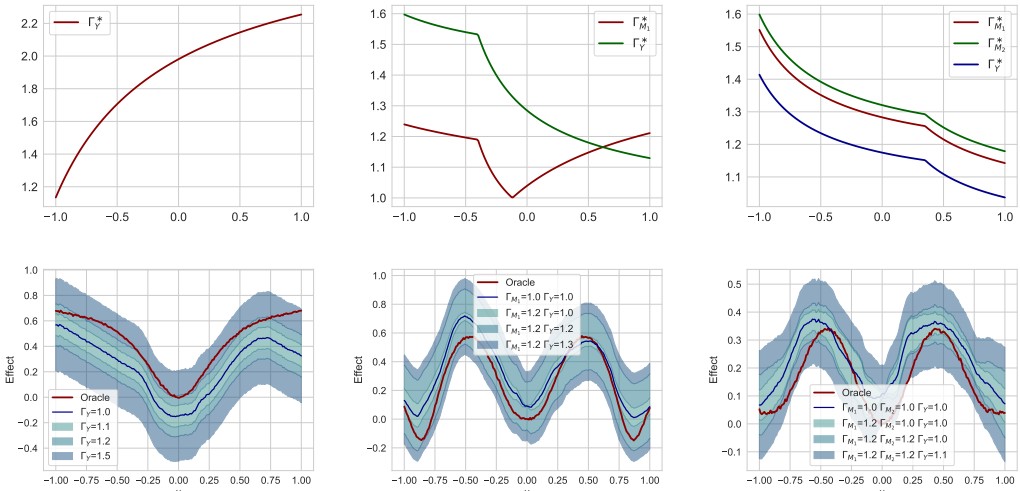

Figure 7: Results for additional treatments in the continuous treatment setting. From left to right is shown: setting (i) with $\bar{a} = 0.9$, setting (ii) with $\bar{a} = (0.2, 0.4)$, and setting (iii) with $\bar{a} = (0.4, 0.5, 0.3)$. The top row shows the oracle sensitivity parameter $\Gamma_W^*$ (depending on $x$), and the bottom row shows the bounds.

treatment decisions twenty times heavier than wrong decisions not to treat. We then compare three different deferral policies: (i) the (standard) policy based on the expectation which treats whenever $\mathbb{E}[Y|\mathbf{X} = \mathbf{x}, A = 1] > \mathbb{E}[Y|\mathbf{X} = \mathbf{x}, A = 0]$, (ii) a more conservative quantile policy that treats whenever $Q_q[Y|\mathbf{X} = \mathbf{x}, A = 1] > Q_{1-q}[Y|\mathbf{X} = \mathbf{x}, A = 0]$ with $q = 0.4$ ($Q_q$ denotes the q-quantile of $\mathbb{P}(Y|\mathbf{X} = \mathbf{x}, A = i)$), and (iii) the same quantile policy with $q = 0.2$. Note that our bounds for policy (i) coincide with sharp bounds for binary CATE from the literature [18]. As done in [31], we report means and standard deviations of 400 random initializations of the IHDP data. The results (Fig. 10) show that the conservative quantile policies improve over the expectation policy.

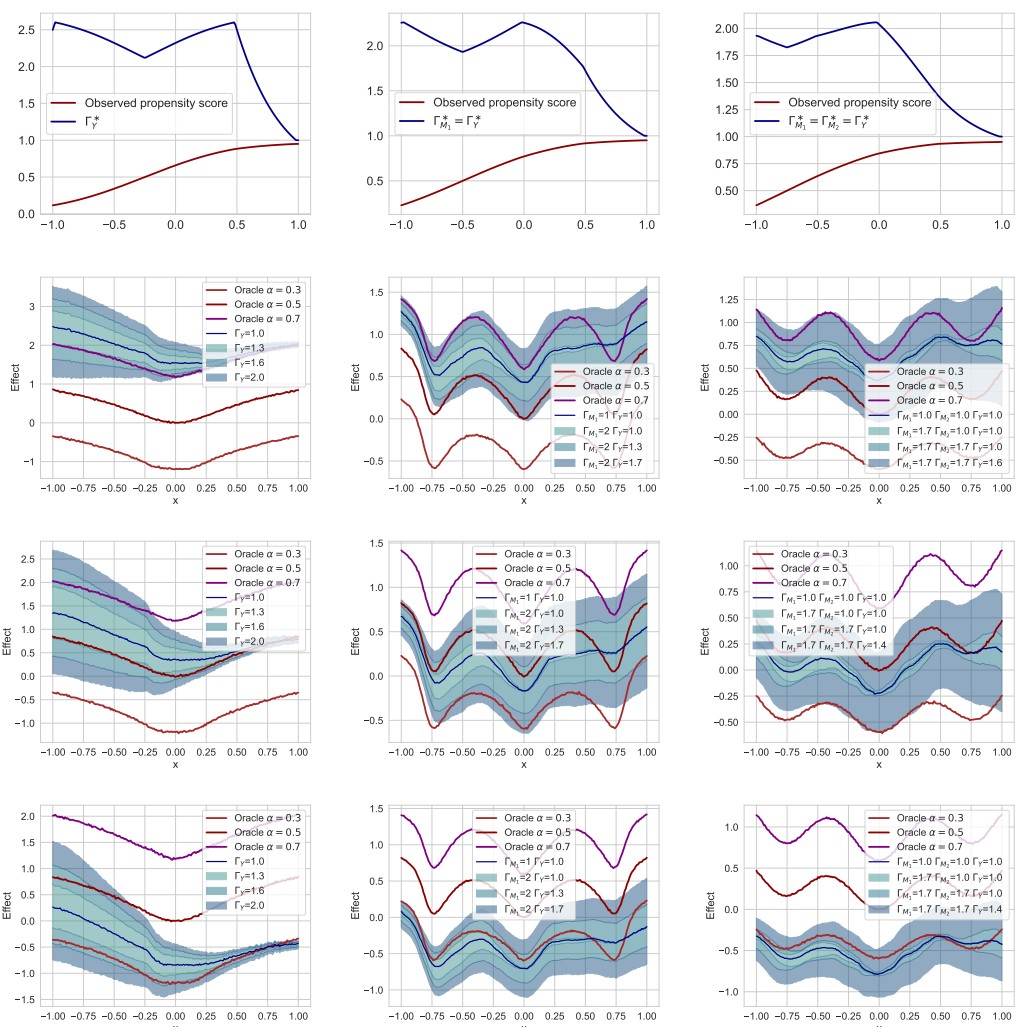

Figure 8: Results for distributional effects in the binary treatment setting using the same treatments as in Fig. 3 (main paper). Settings (i)–(iii) are ordered from left to right. The top row shows the oracle sensitivity parameter $\Gamma_W^*$ (depending on $x$). Rows 2, 3, and 4 show the bounds for the $\alpha$-quantiles of the interventional distribution with $\alpha = 0.7$, $\alpha = 0.5$, and $\alpha = 0.3$.

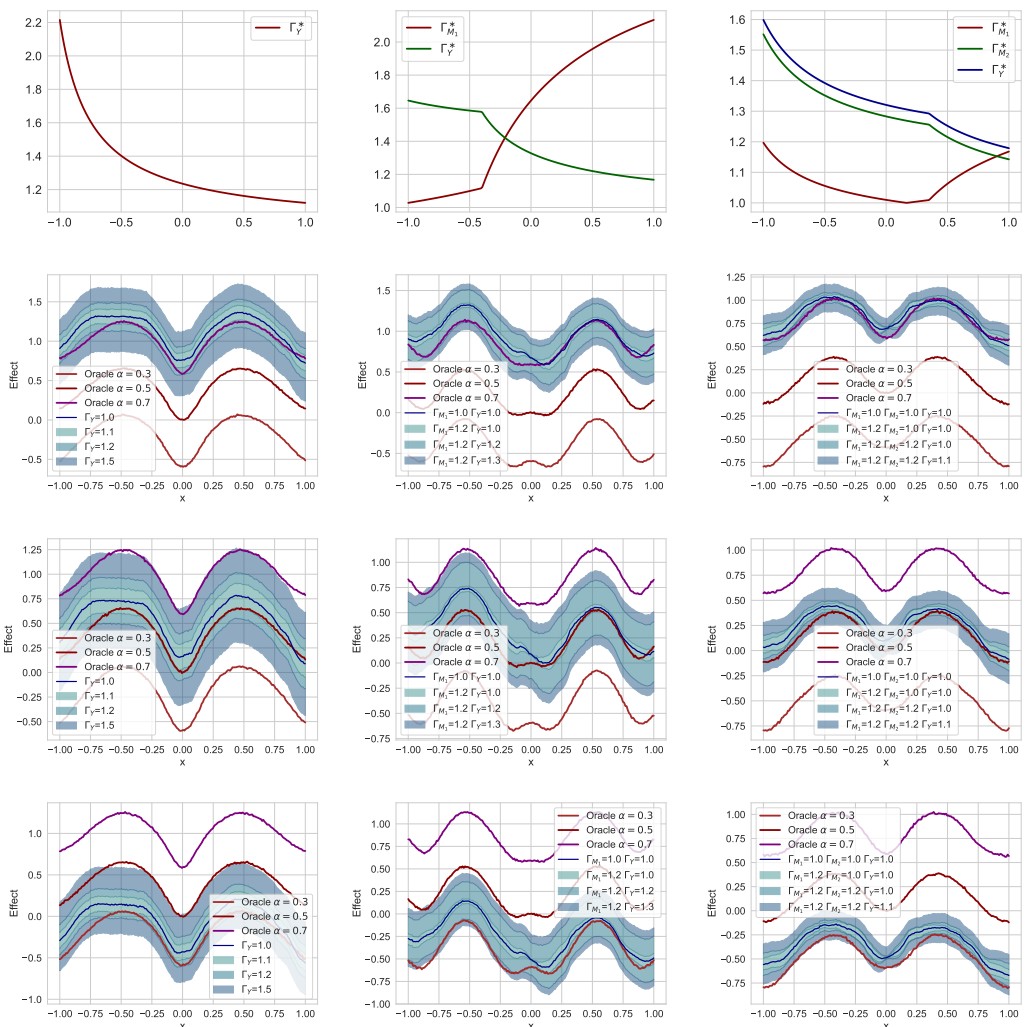

Figure 9: Results for distributional effects in the continuous treatment setting using the same treatments as in Fig. 4 (main paper). Settings (i)–(iii) are ordered from left to right. The top row shows the oracle sensitivity parameter $\Gamma_W^*$ (depending on $x$). Rows 2, 3, and 4 show the bounds for the $\alpha$-quantiles of the interventional distribution with $\alpha = 0.7$, $\alpha = 0.5$, and $\alpha = 0.3$.

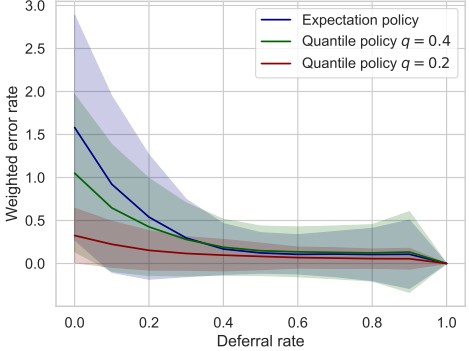

Figure 10: results for the semi-synthetic IHDP data with unobserved confounding from Jesson et al. [31].

