# OpenReview forum: "Sharp Bounds for Generalized Causal Sensitivity Analysis"
_NeurIPS.cc/2023/Conference — NeurIPS 2023 poster_

### Official Review · Reviewer_GZoa · 2023-06-19

**Soundness:** 3 good
**Presentation:** 3 good
**Contribution:** 2 fair
**Rating:** 6
**Confidence:** 4

**Summary:**

The authors generalize a class of causal sensitivity models that includes the traditional MSM, the continuous-treatment CMSM, and the longitudinal (time-varying treatment) LMSM. They show how to compute sharp bounds for the causal estimands by taking inspiration from recent work. Their general framework also allows mediation analysis. They provide an algorithm for computing these bounds.

**Strengths:**

The method is solid and explained well. Figure 2 is nice. Mediation analysis is an important contribution to causal sensitivity models. A broader understanding of sensitivity models is always valuable and the effort to generalize is a good one.

**Weaknesses:**

My main concern is that this generalization that unifies the MSM, CMSM, and LMSM is not very useful. The weighting function seems a bit contrived. It is necessary because the CMSM and LMSM do not use the nominal propensity at all, but the MSM does. Isn't it strange to ignore the nominal propensity and give the same bounds to all potential outcome distributions? Shouldn't the observed confounding inform the unobserved confounding?

A recent alternative to the CMSM is the $\delta$MSM [arXiv:2204.11206] that takes a different approach and appears to perform better. The authors could discuss alternative models like this one or at least keep them in mind when considering general classes of sensitivity models.

The clever approach to sharp partial identification is not novel [see for instance arXiv:2304.10577]. The benefit of the mediation analysis is not really made clear in the results of this submission.


In terms of results, the authors employ a purely synthetic benchmark. The real-world data are interesting but they lack a ground truth. The authors could include some well-known semi-synthetic benchmarks like IHDP and induce hidden confounding by hiding some of the observed confounders.

The authors do not compare with previous methods in their benchmark except for Table 1, where they use a custom weighting function to beat the older CMSM algorithm. That is not very convincing. Also Table 1 should at least be discussed more. The results for the weighted CMSM seem conflicting: tighter bounds but worse coverage?

On a lesser note, the technical details are a bit dense.

**Questions:**

Could you support your reasoning for why this weighting function is a natural interpretation of the more specific sensitivity models? How is it helpful and how can I use it in newer settings? Why should it be set to zero in some cases?

**Limitations:**

Limitations are discussed a bit but they do not address potential societal impacts. It is debatable if that is necessary for this kind of work, but I think pitfalls of these kinds of sensitivity analyses should be discussed.

---

> ### Author Rebuttal · Authors · 2023-08-08
>
> Thank you for your helpful review!
>
> ### Response to “Weaknesses”
>
> * Thank you for your comments. The main motivation behind the GMSM definition (and weighting function) is indeed to unify the three important sensitivity models from the literature.
>     * We argue that our **main contribution** is the derivation of **novel bounds** that are valid for all **sensitivity models**, **distributional effects**, and **mediation/path analysis**. For example, practitioners may use our bounds for the well-established MSM in a mediation analysis setting **without ever explicitly using the GMSM formulation via a weighting function**. Hence, you may simply think about the weighting function as a **notational tool**.
>     * However, we believe that the weighting function has **additional advantages** beyond providing a unified notation when **domain knowledge about the confounding structure is available**. Consider an observational study on the effect of smoking on cancer risk, confounded by certain unobserved genes. It may be known that these genes do not affect the cancer risk for a certain population with covariates $\mathbf{x} \in \mathcal{X}_0$, e.g., female individuals. Hence we can set the weight function to $q(\mathbf{a}, \mathbf{x}) = 1$ for all $\mathbf{x} \in \mathcal{X}_0$ to obtain sharper bounds for the average treatment effect.
>     * **The CMSM sets the weighting function to zero because observed confounding is not necessarily informative for unobserved confounding**. For example, $X$ and $U$ could be independent, or at least nothing about the dependence may be known. The binary MSM uses the propensity score $\mathbb{P}(a \mid x)$ as a weighting function because, for large or small propensities, most of the “randomness” in $A$ is explained by $X$. This implies that $U$ can not have a large effect on $A$ for such $x$. However, this kind of **reasoning breaks down for continuous treatments**. Note that the CMSM has been **proposed in several previous papers** and is not our invention. Furthermore, our paper makes **major contributions in settings beyond continuous treatments**.
>     * **Action:** We will add a detailed discussion about the weighting function to our paper, summarizing the points above.
> * The delta-MSM was **published after the NeurIPS deadline** (UAI 2023), which is why we did not include it in the related work at the time of submission.
>     * We would like to emphasize that it is **not** applicable to **other treatment types**, **mediation/path analysis** or **distributional effects**. Extending our bounds to the delta-MSM may be an interesting direction for future research.
>     * Benchmarking different sensitivity models is difficult due to different assumptions on the data-generating process. We follow **established literature** on sensitivity analysis and study the optimality of bounds under specific sensitivity models, **not the optimality of the sensitivity models** themselves.
>     * **Action:** We will add a discussion to the appendix.
> * The paper mentioned (arXiv:2304.10577) is one of multiple papers that deals with **CATEs for binary treatments**, which we all cite in our related work section. In our paper, **we never claim that our bounds for binary CATE are novel** (e.g., see our abstract). However, our approach and intuition for deriving the bounds **is novel** (via SCM-based probability mass transport, Fig. 2). In contrast to previous work, it can be easily **generalized to more complex causal inference settings** beyond binary CATE. **Action**: We will clarify this in our section on related work.
> * **Mediation analysis** from observational data is crucial in many disciplines such as epidemiology, economics, and algorithmic fairness. It aims to answer questions like “What is the direct causal effect of a medical treatment on health, that is not mediated through a change in diet?”. Our results show that we obtain valid bounds for direct, indirect, and path-specific effects under unobserved confounding. **Action:** We will add a discussion to the appendix.
> * We do not compare against more previous works because: (1) For mediation/path analysis and distributional effects, there **are no baselines** for MSM-based sensitivity analysis. (2) For binary CATE, we obtain the same sharp bounds (in population) as previous work. **Action:** We will add a clarification to the experiment section.
> * With Table 1, we try to convey two messages: (1) Under the CMSM, we get almost the same bounds as Jesson et al. (2022). This is because their algorithm is an **approximation** of our bounds. In contrast, we have **closed-form solutions**, and we thus achieve better **computational speed**. (2) Under a weighted CMSM, we can obtain **tighter bounds** for the average treatment effect, that is, smaller overall interval lengths and lower coverage for small $\Gamma$. Note that we do **not** claim that the weighted CMSM is superior to the unweighted CMSM. As mentioned earlier, we **refrain from benchmarking sensitivity models**. Instead, we outline the possible use of the weight function for incorporating domain knowledge. **Action:** We will clarify this in the paper.
> * **Additional experiments**: Note that we **provided rigorous mathematical proofs for all our results**. However, we agree that adding experiments using IHDP is an excellent idea to improve the experimental section of our paper. \
> **Action:** We performed additional experiments using the IHDP data with hidden confounding from Jesson et al. (2021). Note that IHDP is a dataset with binary treatments and no mediators. Here, our contributions are bounds for **distributional effects**. Details and results are provided in the uploaded PDF file. We also updated our anonymized repository.
> * Thank you for pointing out the density of technical details. **Action:** We will move certain technical details to the appendix.
>
> ### Response to "Limitations"
> * **Action:** As recommended, we will add a discussion on potential societal impacts.

---

> > ### Comment · Reviewer_GZoa · 2023-08-18
> >
> > Thank you for your detailed reply. Your explanations helped me to better understand the framing of sensitivity models using the weighting function. I now have a greater appreciation for the generality of this approach and am correspondingly raising my score to a 6.
> >
> > The reason I am not raising my score further is that I believe the experimental section could have been a bit more fleshed out. The additional IHDP results are interesting, but they appear to only compare distributional/quantile bounds against expectations for a hypothetical downstream task. Since the method proposed in this work generates sharp bounds for a variety of conditions, I would have imagined that the sharpness could be demonstrated in practice to give better bounds than previous approaches, for e.g. partially identifying conditional expectations, in semi-synthetic settings with hidden confounding,

---

> > > ### Author Response · Authors · 2023-08-19
> > > **Thank you for your response and for raising your score**
> > >
> > > Many thanks for acknowledging our rebuttal and for raising your score. Please allow us to elaborate on our experimental results.
> > >
> > > We would like to reiterate that the aim of our paper is not to improve on previous results for binary CATE but rather to generalize existing sharp bounds to other sensitivity models, causal estimands, and causal inference settings. For this purpose, we propose an entirely new approach to deriving sharp bounds in Pearl's SCM framework (see Fig. 2).
> > >
> > > We agree that generally, benchmarking with previous bounds on (semi-)synthetic datasets is desirable to evaluate performance improvement. However, in most settings where our paper has novel contributions (e.g., mediation analysis, distributional effects) **there exist currently no baselines**. **For binary CATE, we obtain exactly the same (sharp) bounds** as Dorn and Guo (2022). That is, the mathematical formulas for the bounds coincide when setting $\mathcal{D} = \mathbb{E}$ for the MSM in our Corollary 1. We believe that the fact that we obtain the same sharp bounds as previous literature is rather encouraging, and indicates (aside from our proofs and experimental results) that our approach for deriving the bounds (Fig. 2) is indeed correct. For continuous CATE, we provide a comparison in Table 1.
> > >
> > > Hence, there is no point in benchmarking our CATE bounds with other approaches for the IHDP data (binary treatment, no mediators). We thus decided to use the IHDP data to illustrate how our bounds for distributional effects can aid decision-making under unobserved confounding.

---

### Official Review · Reviewer_H542 · 2023-07-05

**Soundness:** 3 good
**Presentation:** 2 fair
**Contribution:** 2 fair
**Rating:** 6
**Confidence:** 3

**Summary:**

This paper is about sensitivity analysis (SA) of causal queries in SCMs. In practice, given a causal query and a set of models, the goal is to compute a query's lower and upper bounds. The authors first derive a class of models to be used for SA and show how this extends existing models. An algorithm to obtain the bounds is such cases is derived.

(After the rebuttal, I decided to raise the rating of the paper from 4 to 6)


**Strengths:**

The technical results are sound and non-trivial.
The experiments show good bounds obtained in this way.


**Weaknesses:**

The literature on partially identifiable queries is ignored. In particular existing techniques for bounding such queries are not considered.

**Questions:**

Would it be possible to compare the present method against algorithms for the bounding of non-identifiable queries? E.g. Zhang and Bareinboim, Duarte et al., and Zaffalon et al. worked in this direction in the last two years. I think the sensitivity analysis the authors consider is a heuristic approach to the same problem. If this is true, not having a comparison against these methods is a serious issue.

I also believe that the authors would take advantage of the literature about so-called "imprecise probabilities" and "credal networks" as these models implement the kind of sensitivity analysis of interest for the authors. In particular, the paper "Structural Causal Models Are (Solvable by) Credal Networks" might be a helpful reading.


**Limitations:**

I don't see specific issues in this direction.

---

> ### Author Rebuttal · Authors · 2023-08-08
>
> ## Response to reviewer H542
>
> Thank you for your review and your helpful comments!
>
> ### Response to “Questions”
>
> * Thank you for giving us the opportunity to clarify the difference between two related approaches for bounding causal effects: **Causal sensitivity analysis (CSA)** and **causal partial identification (CPA) without sensitivity models**. In the following, we outline why CSA is **not** a heuristic approach to CPA and why the two approaches should not be benchmarked against each other.
>     * The main difference between CSA and CPA is that CSA imposes **sensitivity models**, that is, assumptions on the “strength” of unobserved confounding, which is controlled by a sensitivity parameter $\Gamma$. Methods for CPA do not make sensitivity assumptions but instead impose other strong assumptions (see points below). In practice, CSA can be used to test the robustness of causal effect estimates to violations of the unconfoundedness assumption (by varying $\Gamma$), while CPA may be applicable in situations where no domain knowledge about the confounding strength is available.
>     * **CSA is not a heuristic approach to CPA.** There is a large stream of literature on CSA in both the machine learning and statistics community (see our related work), with most papers deriving rigorous mathematical results on validity and optimality (“sharpness”) of bounds. In our paper, we provide **formal proofs** for all our results in addition to an extensive empirical evaluation.
>     * In our paper, we study the standard setting for (conditional) average treatment effects with continuous outcomes and optional mediators. Here, CPA is a **special case** of CSA when setting the sensitivity parameter to $\Gamma = \infty$. Hence, CSA provides **provably tighter bounds** than CPA. In fact, CPA bounds are **well characterized** for our setting: For binary treatments, so-called “no assumptions bounds” have been derived in [1]. Indeed, we obtain the same results with our bounds in the limit $\Gamma \to \infty$. For continuous treatments, informative CPA is impossible, that is, the CSA bounds converge to the boundary of the support of the outcome distribution for $\Gamma \to \infty$ [2].
>     * There is no free lunch in causal inference. Because approaches for CPA do not restrict the strength of unobserved confounding, they often impose **other (strong) assumptions** in order to derive informative bounds. For example, all three mentioned papers (Zhang and Bareinboim 2022, Duarte et al. 2021, Zaffalon et al. 2022) are only applicable to settings with **discrete observed variables**. In contrast, these papers are **not** applicable to continuous outcomes, which is the main focus of our paper. Other papers on CPA derive informative bounds by assuming the existence of valid instrumental variables (e.g., [3, 4]).
>     * For the reasons above, there is **no point in benchmarking** our bounds against CPA approaches. In our setting, this would reduce to setting $\Gamma = \infty$. Likewise, methods for only discrete variables or instruments are not applicable, and can thus **not** be used as baselines.
>     * **Summary**: CSA derives bounds under assumptions on the strength of unobserved confounding, and CPA derives bounds under other assumptions and often for more complex/ arbitrary causal graphs (e.g., using valid instrumental variables, or assuming only discrete variables). Both approaches are complementary to each other, and practical usage depends on the assumptions one is willing to impose on the underlying data-generating process.
>     * **Action:** We will expand our related work on CPA (see Appendix A.1) and include the mentioned references. Furthermore, we will add a detailed discussion comparing CSA and CPA, using the points provided above.
> * Thank you for pointing out the literature on imprecise probabilities. It seems to be closely related to the CPA literature as the paper assumes **discrete** **variables** and does **not** impose any **sensitivity models**. We would like to reiterate that, in our setting, CPA is solved and corresponds to setting the sensitivity parameter to $\Gamma = \infty$. However, combining ideas from the imprecise probability literature with CSA could potentially be an interesting direction for future research.  \
> **Action:** We will expand our related work to include the literature on imprecise probabilities, in particular the paper mentioned. We will also add a discussion on potential future work.
>
>
> ### References
>
> [1] Manski 1990, “Nonparametric bounds on treatment effects”, The American Economic Review
>
> [2] Jesson et al. 2022, “Scalable sensitivity and uncertainty analysis for causal-effect estimates of continuous-valued interventions” NeurIPS
>
> [3] Kilbertus et al. 2020, “A Class of Algorithms for General Instrumental Variable Models”, NeurIPS
>
> [4] Padth et al. 2023, “Stochastic Causal Programming for Bounding Treatment Effects”, CLeaR 2023

---

> > ### Comment · Reviewer_H542 · 2023-08-16
> > **Thanks for the clarification on the CSA vs. CPA thing. This motivates me to raise my score.**
> >
> > I appreciate the clarification about the difference between CSA and CPA provided by the authors—many thanks for that. I have experience with CPA but not with CSA. This motivated my question and doubts about the paper. The rebuttal affects my evaluation of the paper, and I am happy to move towards a positive recommendation. Regarding the different points raised by the authors in their rebuttal, I am not very convinced by the argument in the item starting with "There is no free lunch in causal inference". I don't believe that the fact that most of the papers on CPA cope with discrete (endogenous) variables reflects a necessary additional assumption. This is only related to the existing works starting from the more straightforward discrete case. Still, similar techniques will undoubtedly be explored soon also for continuous variables.

---

> > > ### Author Response · Authors · 2023-08-16
> > > **Thank you for your response and raising your score**
> > >
> > > Thank you for acknowledging our response and for raising your score. Please allow us to clarify our argument "There is no free lunch in causal inference". What we meant to say is, that in the standard CATE setting, CPA corresponds to setting $\Gamma \to \infty$ for our bounds. This is because we obtain our bounds by optimizing over all possible SCMs that are compatible with (i) the causal graph, (ii) the observed data distribution, and (iii) the sensitivity constraints. When setting $\Gamma \to \infty$, we ignore (iii) and only constrain our class of SCMs by (i) and (ii), thus performing CPA. **Hence, any other CPA approach for tighter bounds would provably require stronger assumptions.** For example, there are existing CPA approaches that yield tighter bounds by exploiting valid instrumental variables (see e.g., [3, 4]).
> > >
> > > In the following, we characterize our (w.l.o.g. upper) bounds for $\Gamma \to \infty$ in the standard CATE setting: Observed covariates $X$, binary/ continuous treatment $A$, and continuous outcome $Y$. We are interested in the causal query $Q(x, a, \mathcal{M}) = \mathbb{E}\left[Y \mid x, do(A = a)\right]$ (Example 1 from our paper). Our upper bound is $Q^+ = \int_{\ell}^{F^{-1}(c_Y^+)} \frac{y}{s_Y^+} \mathbb{P}(y \mid x, a) dy +  \int_{F^{-1}(c_Y^+)}^{u} \frac{y}{s_Y^-} \mathbb{P}(y \mid x, a) dy $, where $c_Y^+ = \frac{\Gamma}{1 + \Gamma}$ and $\ell, u$ are the lower/ upper support points of the distribution $\mathbb{P}(y \mid x, a)$, respectively.
> > >
> > > 1) For binary treatment $A \in \{0, 1\}$, using the MSM we obtain
> > > \begin{equation}
> > > Q^+ = \int_{\ell}^{F^{-1}(\frac{\Gamma}{1 + \Gamma})} y \left((1 - \Gamma^{-1}) \mathbb{P}(a \mid x) + \Gamma^{-1} \right) \mathbb{P}(y \mid x, a) dy +  \int_{F^{-1}(\frac{\Gamma}{1 + \Gamma})}^{u} y \left((1 - \Gamma) \mathbb{P}(a \mid x) + \Gamma \right) \mathbb{P}(y \mid x, a) dy  \xrightarrow[\Gamma \to \infty]{} \mathbb{P}(a \mid x) \mathbb{E}[Y \mid x, a] + (1 - \mathbb{P}(a \mid x)) u
> > > \end{equation}
> > > This bound is also known as the "no assumptions bound" or "Manski bound", originally derived in [1]. Of note, with our theory, we can derive similar bounds for distributional effects. Hence, **our paper even makes non-trivial contributions to CPA**. We will add this to our paper.
> > >
> > > 2) For continuous treatments, using the CMSM we obtain
> > > \begin{equation}
> > > Q^+ = \int_{\ell}^{F^{-1}(\frac{\Gamma}{1 + \Gamma})} \frac{y}{\Gamma} \mathbb{P}(y \mid x, a) dy +  \int_{F^{-1}(\frac{\Gamma}{1 + \Gamma})}^{u} y \Gamma \mathbb{P}(y \mid x, a) dy  \xrightarrow[\Gamma \to \infty]{} u
> > > \end{equation}
> > > That is, for continuous treatments the corresponding "no assumptions bound" is exactly the right support point of the observed distribution (see also [2]). Hence, informative CPA for continuous treatments and continuous outcomes is **not possible without imposing stronger assumptions** (such as IVs).
> > >
> > > **Action**: We will add the derivations above to our discussion on CSA vs CPA.

---

### Official Review · Reviewer_taFV · 2023-07-05

**Soundness:** 3 good
**Presentation:** 4 excellent
**Contribution:** 4 excellent
**Rating:** 8
**Confidence:** 3

**Summary:**

The authors propose a unified framework for causal sensiitivity analysis under unobserved confounding that generalizes the Marginal Sensitivity Model (MSM). They derive sharp bounds for a diverse set of causal effects such as the CATE, mediation and path analysis effects, and distributional effects. The framework is applicable to discrete, continuous, and time-varying interventions. They offer, to my knowledge, a novel interpretation of the marginal sensitivity model via SCM. They show that in the case of binary treatments, their derived bounds  coincide with the optimality result of Dorn and Guo 2023. They provide a closed form solution to and algorithm to estimate the bounds and show empircally that it improves over line search methods (a formal complexity analysis would be nice).

**Strengths:**

This paper makes several novel contributions to an active area of study in causal machine learning, which are listed above in the summary. They provide theoretical and experimental evidence supporting their claims. They do a good job presenting and comparing to the related work. An exceptionally well writen and organized paper given the complexity of the subject matter.

**Weaknesses:**

## I have one primary comment.

I think there may be a step missing in the special cases proofs of Appendix C. I appreciate how the authors have utilized SCM to define the GMSM. But, in defining the  MSM, CMSM, and LMSM in terms of hidden confounders $u$, it seems that there is a step missing from how these models are originally defined. Namely, they are defined with respect to potential-outcomes / counterfactuals $Y_{t}$ and the conditional independence relation: $Y_t \perp T \mid X$. For example, it's not obvious to me how you move from $P(a \mid x, y_t)$, to $P(a \mid x, u)$. Given that you show equivalence in terms of $P(a \mid x, u)$, I think it is important to be explicit here. If this is resolved, and I admit that this could be completely trivial and I just don't see it, I would happily increase my score. If it cannot be resolved, I would suggest removing these claims and my score would remain the same.

## I have a few minor comments.

First, the last two paragraphs of the introduction can be streamlined as the contributions paragraph essentially reiterates the points of the paragraph starting on line 45. I like both styles, with slight preference for the contribution format.

There is a recent paper proposing a marginal sensitivity analysis for continuous treatments that could be added to the related works.

Line 70 "... when while ..." seems to be a typo

**Questions:**

How does one show that  $P(a \mid x, y_t) = P(a \mid x, u)$?

**Limitations:**

Yes

---

> ### Author Rebuttal · Authors · 2023-08-08
>
> ## Response to reviewer taFV
>
> Thank you for your positive evaluation of our paper! We took all your comments at heart and improved our paper accordingly.
>
> ### Response to “A formal complexity analysis would be nice”
>
> * Thank you for pointing this out. In the following, we assume that all models are trained and we have a sample $(y_i)_{i=1}^k \sim \hat{\mathbb{P}}^Y(\cdot \mid \mathbf{x}, \mathbf{m}, \mathbf{a})$ (necessary for both estimators) available. Then, our estimator from Eq. (8) has a complexity of $\mathcal{O}(k)$ because it only involves summing and quantile computation. Algorithm 1 of Jesson et al. (2022) has a complexity of  $\mathcal{O}(k n)$, where $n$ is the number of grid search points. For Table 1, we choose $k = n = 5000$.  \
> **Action:** We will add this to our appendix and thereby highlight the benefits our work.
>
> ### Response to “Primary comment”
>
> * This is an excellent question. First of all, while we are the first to use Pearl’s SCM framework for MSM-type sensitivity analysis, we are **not** the first to define the MSM in terms of an unobserved confounder $U$ instead of the potential outcomes $Y_t$. Note that the SCM and potential outcome framework are logically equivalent and we decided to use SCMs because they allowed us to think in a new way about the bounding problem for sensitivity analysis (via probability mass transport, see Fig. 2). The following state-of-the-art papers all use the unobserved confounder $U$ in their definitions within the potential outcomes framework:
>     * [1] Dorn and Guo 2022, “Sharp sensitivity analysis for inverse propensity weighting via quantile balancing”, JASA, Equation (2)
>     * [2] Dorn et al. 2022, “Doubly-valid/ doubly-sharp sensitivity analysis for causal inference with unmeasured confounding”, arXiv:2112.11449, Definition 1
>     * [3] Oprescu et al. 2023, “B-learner: Quasi-oracle bounds on heterogeneous causal effects under hidden confounding”, ICML, Assumption 1
>     * [4] Bonvini and Kennedy 2022, “Sensitivity analysis for marginal structural models”, arXiv:2210.04681, Equation (5)
> * We agree that the equivalence of the two MSM definitions is not immediately clear and that the papers above do not seem to provide formal results on this. In [1], the authors only write “However, as pointed out by a referee, these assumptions are equivalent”. In the following, we provide a formal result using the GMSM formulation (note that this is equivalent to the original MSM definition via log-odds ratio).
> * Lemma: The following two statements are equivalent:
>     1) There exist a $U$ with $Y_1, Y_0 \perp  A \mid X, U$ so that $s^- \leq \frac{\mathbb{P}(u \mid x, a)}{\mathbb{P}(u \mid x)} \leq s^+$
>     2) It holds that $s^- \leq \frac{\mathbb{P}(y_1, y_0 \mid x, a)}{\mathbb{P}(y_1, y_0 \mid x)} \leq s^+$
> * Proof (sketch):
>     * Direction 2. $\to$ 1.: Define $U = (Y_1, Y_0)$.
>     * Direction 1. $\to$ 2.: We proceed via proof by contradiction. Assume there exists a pair $(y_1, y_0)$ that violates 2), say w.l.o.g. $\frac{\mathbb{P}(y_1, y_0 \mid x, a)}{\mathbb{P}(y_1, y_0 \mid x)} > s^+$. We can use the ignorability condition from 1. to write $\mathbb{P}(y_1, y_0 \mid x, a) = \int \mathbb{P}(y_1, y_0 \mid x, u, a) \mathbb{P}(u \mid x, a) du = \int \mathbb{P}(y_1, y_0 \mid x, u) \mathbb{P}(u \mid x, a) du $. Furthermore, $\mathbb{P}(y_1, y_0 \mid x) = \int \mathbb{P}(y_1, y_0 \mid x, u) \mathbb{P}(u \mid x) du$. It follows that $s^+ &lt; \frac{\int \mathbb{P}(y_1, y_0 \mid x, u) \mathbb{P}(u \mid x, a) du}{\int \mathbb{P}(y_1, y_0 \mid x, u) \mathbb{P}(u \mid x) du} $. Hence, there exists a $u$ such that $s^+ &lt; \frac{\mathbb{P}(y_1, y_0 \mid x, u) \mathbb{P}(u \mid x, a)}{\mathbb{P}(y_1, y_0 \mid x, u) \mathbb{P}(u \mid x)} = \frac{\mathbb{P}(u \mid x, a)}{\mathbb{P}(u \mid x)}$, which is a contradiction to the sensitivity constraint from 1.
> * **Action**: We will add a section to the appendix where we discuss the two definitions and prove their equivalence with the arguments provided above.
>
> ### Response to “Minor comments”
>
> * Thank you for pointing out the redundancy. **Action:** We will streamline the last two paragraphs of the introduction into a larger “Contribution” paragraph, as proposed.
> * We assume that you are referring to the following paper: “Partial identification of dose responses with hidden confounders”, Marmarelis et al., UAI 2023. Thanks for pointing this out! The paper was published after the NeurIPS deadline, which is why we did not include it in the related work at the time of submission. We would like to emphasize that the paper derives bounds for a different sensitivity model that is only applicable to continuous treatments and **not** to binary or time-varying treatments, and mediation or distributional effects. **Action:** We will add the paper to the related work and also add a discussion to the appendix, in which we compare the different sensitivity models for continuous treatments.
> * Thanks for pointing out the typo! **Action:** We will carefully proofread our paper.

---

> > ### Comment · Reviewer_taFV · 2023-08-11
> > **Thank you for your response. You have addressed my concerns and I would like to raise my score to an 8.**
> >
> > I have read your response and I appreciate the efforts you have made to address my concerns. I'm particularly impressed by the lemma you have provided, as the relationship there is something that has bothered me for some time. Trusting that the action points will be incorporated into the camera ready version, I would like to increase my score to an 8.

---

> > > ### Author Response · Authors · 2023-08-14
> > >
> > > Thank you for acknowledging our response and for providing swift feedback. We are happy that we were able to resolve the ambiguity regarding the sensitivity model definition. We will incorporate all action points as promised.
> > >
> > > Thank you also for your willingness to increase your score to an 8. We saw in the system that the number did not change yet, and, for that reason, we wanted to simply follow back if this is still a to-do or if this is something with OpenReview. If you have any further questions or requests, please let us know.

---

### Official Review · Reviewer_9HdS · 2023-07-06

**Soundness:** 4 excellent
**Presentation:** 3 good
**Contribution:** 3 good
**Rating:** 6
**Confidence:** 4

**Summary:**

The authors study the problem of bounding a given causal effect. To this end, they propose a generalized marginal sensitivity model (GMSM) that is applicable to multiple discrete, continuous , and time-varying treatments. They also present a new interpretation of the partial identification.

**Strengths:**

The proposed GMSM model generalizes the previous models and leads to sharp bound for certain causal effects with certain causal graphs.

The bounds depend on observed variables and thus can be estimated.

Except some minor ambiguities (see below), the paper is well-written.

**Weaknesses:**

Although the proposed model generalized the previous models but the presented theoretical results hold under specific  graphical constraints e.g., no confounder between M and outcome, no hidden confounders between X and $\{A,M,Y\}$.

The other weakness about the results is its generalizability to arbitrary causal graphs. Based on the presented proofs in the appendix, it is not clear how this results can be generalized by relaxing the assumptions.

There is ambiguity about the notation of U. Does $U_Y$ denote the unobserved exogenous variable for Y in the definition of SCM or is it a hidden confounder between Y and A? Is it possible to have hidden confounders among the mediators (e.g., $M_1$ and $M_2$)?

The explanation below (3) says  “If U_w has no effect on A, Eq. (3) holds with $s^-_W (a, x) = S^+_W (a, x) = 1$”. But it seems that (3) encodes the effect of A on $U_W$.!

In (35) in the Appendix, what is U exactly?  Does the setting imply that u and x are independent, i.e., $p(u|x)=p(u)$?

**Questions:**

See above.

**Limitations:**

 The authors addressed the limitations.

---

> ### Author Rebuttal · Authors · 2023-08-08
>
> ## Response to reviewer 9HdS
>
> Thank you for your positive review and your helpful comments. We improved our paper in the following ways:
>
> ### Response to “Weaknesses”
>
> * Thank you for giving us the opportunity to clarify the assumptions in our paper.
>     * The assumption that no unobserved confounders exist between mediators $M$ and outcome $Y$ is necessary to provide results for **nested counterfactuals** (see point three below for a more detailed discussion). We would like to emphasize that we need this assumption only for settings with mediators (i.e., Corollary 2) and **not** for our results without mediators (Corollary 1). Furthermore, this assumption is **weaker** than assumptions usually taken in the mediation literature (see point three).
>     * We do **not** assume that there is no unobserved confounding between $X$ and $A, M, Y$.  For example, we allow for correlation between $X$ and $U$, so that $U$ could also be an unobserved confounder between $X$ and $Y$. Technically speaking, we only assume that $\{X, U_W\}$ is a sufficient adjustment set for the $A-W$ relationship ($W \in \{M, Y\}$). This is equivalent to ignorability assumptions in the potential outcomes framework like $Y_1, Y_0 \perp A \mid X, U$ (for binary treatments), which are **standard in the literature** on unobserved confounding and imply that $U$ captures all unobserved confounders (Dorn and Guo 2022).
>     * **Action:** We will add a detailed discussion of our assumptions to our paper.
>
> * We agree that our results cannot be generalized for arbitrary causal graphs in a straightforward way. However, the aim of our paper is not to provide results for arbitrary graphs, but rather for (conditional) treatment effects and mediation/ path analysis. We argue that the setting we study in our paper is already **very general compared to current state-of-the-art** work on causal sensitivity analysis. In particular, we are the first to study MSM-type sensitivity analysis for mediation and path analysis. Furthermore, our setting includes the whole literature on (conditional) average treatment effect estimation as a special case. We believe that extending our results to other causal graphs (e.g., with instrumental variables) is an interesting direction of possible future research, but out of the scope of this paper.  \
> **Action:** We will expand our section on future work and limitations, and discuss possible causal inference settings which may be of interest for future work on sensitivity analysis.
>
> * Thank you for pointing out the ambiguity in our notation $U_W$. All variables $U_W$ we consider explicitly are unobserved confounders between treatment $A$ and mediator/ outcome $W$ and not exogenous noise. The latter is implicitly considered as part of the SCM definition.  In Theorem 1, we assume that there exist no unobserved confounders between mediators, e.g., between $M_1$ and $M_2$.
>     * The main reason for this assumption is that it allows us to interpret the causal query $Q(\mathbf{x}, \overline{\mathbf{a}}, \mathcal{M})$ from Equation (1) as a path-specific effect. Path-specific effects are defined as so-called nested counterfactuals that lie in the third layer of Pearl’s causal hierarchy. Using the assumption from Theorem 1, they can be reduced to the query $Q$, which lies in layer 2 of Pearl’s hierarchy, i.e., only depends on interventions (Correa et al. 2021). Our sensitivity analysis then bridges the gap from layer 2 to layer 1 (observational data). We believe that, in principle, relaxing the assumption should be possible. For example, one could consider combining our results with a sensitivity analysis from layer 3 to layer 2, e.g., by imposing a sensitivity model on the level of confounding between mediators $M_1$ and $M_2$. While this goes beyond the scope of our paper, it seems like an interesting direction for future work.
>     * Note that **we rely on this assumption only in mediation settings**. A large part of our contribution is the derivation of bounds for distributional effects for different treatment types in settings without mediators (Corollary 1), where we do not impose such an assumption.
>     * We would like to emphasize that our assumptions are **weaker** than most state-of-the-art work for mediation/path analysis (e.g., Shpister and Tchetgen Tchetgen 2016). Most work assumes unconfoundedness between all variables, while we allow for unobserved confounding between treatment and mediators/ outcome and only assume unconfoundedness between the mediators and outcome.
>     * **Action:** We expand our discussion of the assumption from Theorem 1, clarifying that we do not allow for unobserved confounding between mediators. We also expand our section on future work and clarify the definition of $U_W$.
>
> * In Eq. (3), $U_W$ is by definition a parent of $A$ in the causal graph. Eq. (3) only limits the level of dependence (in the language of probability theory) between the random variables $U_W$ and $A$. Here, the order of the variables in $\mathbb{P}(u_W \mid x, a)$ does not reflect the causal order between $U_W$ and $A$. Note also that for the MSM we can simply rewrite the fraction from Eq. (3) as $\frac{\mathbb{P}(a \mid x, u_W)}{\mathbb{P}(a \mid x)}$ (see Appendix C), which reverses the order of the variables in the probabilities.  \
> **Action:** We will add a clarification to the paper.
>
> * Eq. (35) is part of Appendix C, where we consider the three important sensitivity models MSM, CMSM, and LMSM. These sensitivity models are defined without mediators, which means that we only have one unobserved confounder $U = U_Y$ between treatment $A$ and outcome $Y$. We do **not** assume that U and X are independent, as this would be a very strong assumption. For example, if we are interested in the effect of smoking ($A$) on cancer risk ($Y$), $U$ may be specific gene expressions correlated with observed confounders $X$ such as medical history. \
> **Action:** We will add a clarification to Appendix C.

---

### Author Rebuttal · Authors · 2023-08-08

## Response to all reviewers

Thank you very much for the constructive evaluation of our paper and your helpful comments! We addressed all of them in the comments below and uploaded additional empirical results as a PDF file. Our main improvements are the following:



* We provided **clarifications** regarding our assumptions. Therein, we explain that our assumptions are weaker than in previous literature.
* We discussed the **connection to related literature** (e.g., partial identification without sensitivity models) and spell out explicitly how our work is different and novel.
* We obtained **new empirical and theoretical results**, including a new experiment on semi-synthetic data.

We will incorporate all changes (labeled with **Action)** into the camera-ready version of our paper. Given these improvements, we are confident that our paper will be a valuable contribution to the causal machine learning literature and a good fit for NeurIPS 2023.

---

### Comment · Area_Chair_vLL7 · 2023-08-18
**Acknowledging author rebuttals**

Dear all,

I want to thank the authors for their rebuttals and want to acknowledge that these will be taken into account.
Unfortunately, 2 reviewers have still not replied to the author rebuttal and I explicitly urge them (again) to please reply to the rebuttals as soon as possible since the author/reviewer discussion period ends soon.

There are still substantial discrepancies in the scores this submission has received so far and the reviewer-author discussion is crucial to clarify the different viewpoints.

Best regards

---

### Decision · Program_Chairs · 2023-09-21

**Decision:**

Accept (poster)

**Comment:**

All reviewers agreed about the novelty and relevance of this work to the NeurIPS community. The authors managed to iron out a few initial misunderstandings and unclear points during the rebuttal and we strongly encourage them to include the promised action points (also new theoretical results developed during the rebuttal, etc.) in the final version.